# Light Unbalanced Optimal Transport

**Milena Gazdieva**
Skolkovo Institute of Science and Technology
Artificial Intelligence Research Institute
Moscow, Russia
`milena.gazdieva@skoltech.ru`

**Arip Asadulaev**
ITMO University
Artificial Intelligence Research Institute
Moscow, Russia
`asadulaev@airi.net`

**Evgeny Burnaev**
Skolkovo Institute of Science and Technology
Artificial Intelligence Research Institute
Moscow, Russia
`e.burnaev@skoltech.ru`

**Alexander Korotin**
Skolkovo Institute of Science and Technology
Artificial Intelligence Research Institute
Moscow, Russia
`a.korotin@skoltech.ru`

## Abstract

While the continuous Entropic Optimal Transport (EOT) field has been actively developing in recent years, it became evident that the classic EOT problem is prone to different issues like the sensitivity to outliers and imbalance of classes in the source and target measures. This fact inspired the development of solvers that deal with the *unbalanced* EOT (UEOT) problem − the generalization of EOT allowing for mitigating the mentioned issues by relaxing the marginal constraints. Surprisingly, it turns out that the existing solvers are either based on heuristic principles or heavy-weighted with complex optimization objectives involving several neural networks. We address this challenge and propose a novel theoretically-justified, lightweight, unbalanced EOT solver. Our advancement consists of developing a novel view on the optimization of the UEOT problem yielding tractable and a non-minimax optimization objective. We show that combined with a light parametrization recently proposed in the field our objective leads to a fast, simple, and effective solver which allows solving the continuous UEOT problem in minutes on CPU. We prove that our solver provides a universal approximation of UEOT solutions and obtain its generalization bounds. We give illustrative examples of the solver's performance. The code is publicly available at

https://github.com/milenagazdieva/LightUnbalancedOptimalTransport

## 1 Introduction

The computational *optimal transport* (OT) has proven to be a powerful tool for solving various popular tasks, e.g., image-to-image translation [68, 17, 50, 28], image generation [67, 13, 7] and biological data transfer [5, 42, 66]. Historically, the majority of early works in the field were built upon solving the OT problem between discrete probability measures [10, 53]. Only recently the advances in the field of generative models have led to explosive interest from the ML community in developing the **continuous** OT solvers, see [38] for a survey. The setup of this problem assumes that the learner needs to estimate the OT plan between continuous measures given only empirical samples of data from them. Due to convexity-related issues of OT problem [40], many works consider the EOT problem, i.e., use *entropy* regularizers which guarantee, e.g., the uniqueness of learned plans.

Meanwhile, researches attract attention to other shortcomings of the classic OT problem. It enforces hard constraints on the marginal measures and, thus, does not allow for mass variations. As a result, OT shows high sensitivity to an imbalance of classes and outliers in the source and target

38th Conference on Neural Information Processing Systems (NeurIPS 2024).

measures [4] which are almost inevitable for large-scale datasets. To overcome these issues, it is common to consider extensions of the OT problem, e.g., unbalanced OT/EOT (UOT/UEOT) [8, 43]. The unbalanced OT/EOT formulations allow for variation of total mass by relaxing the marginal constraints through the use of divergences.

The scope of our paper is the continuous UEOT problem. It seems that in this field, a solver that is fast, light, and theoretically justified has not yet been developed. Indeed, many of the existing solvers follow a kind of heuristical principles and are based on the solutions of discrete OT. For example, [45] uses a regression to interpolate the discrete solutions, and [16, 36] build a flow matching upon them. Almost all of the other solvers [9, 70] employ several neural networks with many hyper-parameters and require time-consuming optimization procedures. We solve the aforementioned shortcomings by introducing a novel lightweight solver that can play the role of a simple baseline for unbalanced EOT.

**Contributions.** We develop a novel *lightweight* solver to estimate continuous **unbalanced** EOT couplings between probability measures (§4). Our solver has a non-minimax optimization objective and employs the Gaussian mixture parametrization for the UEOT plans. We provide the generalization bounds for our solver (§4.4) and experimentally test it on several tasks (§5.1, §5.2).

**Notations.** We work in the Euclidian space $(\mathbb{R}^d, \|\cdot\|)$. We use $\mathcal{P}_{2,ac}(\mathbb{R}^d)$ to denote the set of absolutely continuous (a.c.) Borel probability measures on $\mathbb{R}^d$ with finite second moment and differential entropy. The set of nonnegative measures on $\mathbb{R}^d$ with finite second moment is denoted as $\mathcal{M}_{2,+}(\mathbb{R}^d)$. We use $\mathcal{C}_2(\mathbb{R}^d)$ to denote the space of all *continuous* functions $\zeta : \mathbb{R}^d \to \mathbb{R}$ for which $\exists a = a(\zeta), b = b(\zeta)$ such that $\forall x \in \mathbb{R}^d : |\zeta(x)| \leq a + b\|x\|^2$. Its subspace of functions which are additionally *bounded from above* is denoted as $\mathcal{C}_{2,b}(\mathbb{R}^d)$. For a.c. measure $p \in \mathcal{P}_{2,ac}(\mathbb{R}^d)$ (or $\mathcal{M}_{2,+}(\mathbb{R}^d)$), we use $p(x)$ to denote its density at a point $x \in \mathbb{R}^d$. For a given a.c. measure $\gamma \in \mathcal{M}_{2,+}(\mathbb{R}^d \times \mathbb{R}^d)$, we denote its total mass by $\|\gamma\|_1 \stackrel{\text{def}}{=} \int_{\mathbb{R}^d \times \mathbb{R}^d} \gamma(x, y)dxdy$. We use $\gamma_x(x), \gamma_y(y)$ to denote the marginals of $\gamma(x, y)$. They satisfy the equality $\|\gamma_x\|_1 = \|\gamma_y\|_1 = \|\gamma\|_1$. We write $\gamma(\cdot|x)$ to denote the conditional *probability* measure. Each such measure has a unit total mass. We use $\overline{f}$ to denote the Fenchel conjugate of a function $f$: $\overline{f}(t) \stackrel{\text{def}}{=} \sup_{u \in \mathbb{R}}\{ut - f(u)\}$. We use $\mathbb{I}_A$ to denote the convex indicator of a set $A$, i.e., $\mathbb{I}_A(x) = 0$ if $x \in A$; $\mathbb{I}_A(x) = +\infty$ if $x \notin A$.

## 2 Background

Here we give an overview of the relevant entropic optimal transport (EOT) concepts. For additional details on balanced EOT, we refer to [10, 24, 53], unbalanced EOT - to [8, 43].

$f$-**divergences for positive measures**. For *positive* measures $\mu_1, \mu_2 \in \mathcal{M}_{2,+}(\mathbb{R}^{d'})$ and a lower semi-continuous function $f : \mathbb{R} \to \mathbb{R} \cup \{\infty\}$, the *f-divergence* between $\mu_1, \mu_2$ is defined by

$$D_f(\mu_1\|\mu_2) \stackrel{\text{def}}{=} \int_{\mathbb{R}^{d'}} f\left(\frac{\mu_1(x)}{\mu_2(x)}\right)\mu_2(x)dx \text{ if } \mu_1 \ll \mu_2 \text{ and } +\infty \text{ otherwise.}$$

We consider $f(t)$ which are convex, non-negative and attain zero uniquely when $t = 1$. In this case, $D_f$ is a valid measure of dissimilarity between two positive measures (see Appendix C for details). This means that $D_f(\mu_1\|\mu_2) \geq 0$ and $D_f(\mu_1\|\mu_2) = 0$ if and only if $\mu_1 = \mu_2$. In our paper, we also assume that all $f$ that we consider satisfy the property that $\overline{f}$ is a *non-decreasing* function.

Kullback-Leibler divergence $D_{\text{KL}}$ [8, 62], is a particular case of such $f$-divergence for positive measures. It has a generator function $f_{\text{KL}}(t) \stackrel{\text{def}}{=} t\log t - t + 1$. Its convex conjugate $\overline{f_{\text{KL}}}(t) = \exp(t) - 1$. Another example is the $\chi^2$-divergence $D_{\chi^2}$ which is generated by $f_{\chi^2}(t) \stackrel{\text{def}}{=} (t-1)^2$ when $t \geq 0$ and $\infty$ otherwise. The convex conjugate of this function is $\overline{f_{\chi^2}}(t) = -1$ if $t < -2$ and $\frac{1}{4}t^2 + t$ when $t \geq 2$.

*Remark.* By the definition, convex conjugates of $f_{\text{KL}}$ and $f_{\chi^2}$ divergences are proper, non-negative and non-decreasing. These properties are used in some of our theoretical results.

**Entropy for positive measures.** For $\mu \in \mathcal{M}_{2,+}(\mathbb{R}^{d'})$, its entropy [8] is given by

$$H(\mu) \stackrel{\text{def}}{=} -\int_{\mathbb{R}^{d'}} \mu(x)\log\mu(x)dx + \|\mu\|_1, \text{ if } \mu \text{ is a.c. and } -\infty \text{ otherwise.} \quad (1)$$

When $\mu \in \mathcal{P}_{2,ac}(\mathbb{R}^{d'})$, i.e., $\|\mu\|_1 = 1$, equation (1) is the usual differential entropy minus 1.

**Classic EOT formulation (with the quadratic cost).** Consider two probability measures $p \in \mathcal{P}_{2,ac}(\mathbb{R}^d)$, $q \in \mathcal{P}_{2,ac}(\mathbb{R}^d)$. For $\varepsilon > 0$, the EOT problem between $p$ and $q$ is

$$\min_{\pi \in \Pi(p,q)} \int_{\mathbb{R}^d} \int_{\mathbb{R}^d} \frac{\|x-y\|^2}{2} \pi(x,y)dxdy - \varepsilon H(\pi), \tag{2}$$

where $\Pi(p,q)$ is the set of probability measures $\pi \in \mathcal{P}_{2,ac}(\mathbb{R}^d \times \mathbb{R}^d)$ with marginals $p$, $q$ (transport plans). Plan $\pi^*$ attaining the minimum exists, it is unique and called the *EOT plan.*

Classic EOT imposes hard constraints on the marginals which leads to several issues, e.g., sensitivity to outliers [4], inability to handle potential measure shifts such as class imbalances in the measures $p, q$. The UEOT problem [70, 9] overcomes these issues by relaxing the marginal constraints [62].

**Unbalanced EOT formulation (with the quadratic cost).** Let $D_{f_1}$ and $D_{f_2}$ be two $f$-divergences over $\mathbb{R}^d$. For two probability measures $p \in \mathcal{P}_{2,ac}(\mathbb{R}^d)$, $q \in \mathcal{P}_{2,ac}(\mathbb{R}^d)$ and $\varepsilon > 0$, the unbalanced EOT problem between $p$ and $q$ consists of finding a minimizer of

$$\inf_{\gamma \in \mathcal{M}_{2,+}(\mathbb{R}^d \times \mathbb{R}^d)} \int_{\mathbb{R}^d} \int_{\mathbb{R}^d} \frac{\|x-y\|^2}{2} \gamma(x,y)dxdy - \varepsilon H(\gamma) + D_{f_1}(\gamma_x\|p) + D_{f_2}(\gamma_y\|q), \tag{3}$$

see Fig. 1. Here the minimum is attained for a unique $\gamma^*$ which is called the *unbalanced optimal entropic* (UEOT) plan. Typical choices of $f_i$ ($i \in [1,2]$) are $f_i(t) = \tau_i f_{\text{KL}}(t)$ or $f_i(t) = \tau_i f_{\chi^2}(t)$ ($\tau_i > 0$) yielding the scaled $D_{\text{KL}}$ and $D_{\chi^2}$, respectively. In this case, the bigger $\tau_1$ ($\tau_2$) is, the more $\gamma_x$ ($\gamma_y$) is penalized for not matching the corresponding marginal distribution $p$ ($q$).

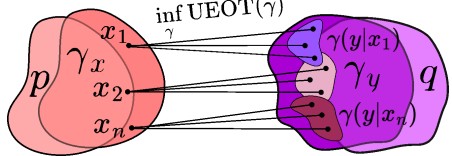

Figure 1: Unbalanced EOT problem.

*Remark 1.* While the set $\mathcal{M}_{2,+}(\mathbb{R}^d \times \mathbb{R}^d)$ contains not only a.c. measures, infimum in problem (3) is attained for a.c. measure $\gamma^*$ since $-\varepsilon H(\gamma^*)$ term turns to $+\infty$ otherwise. Thus, almost everywhere in the paper we are actually interested in the subset of a.c. measures in $\mathcal{M}_{2,+}(\mathbb{R}^d \times \mathbb{R}^d)$.

*Remark 2.* The balanced EOT problem (2) is a special case of (3). Indeed, let $f_1$ and $f_2$ be the convex indicators of $\{1\}$, i.e., $f_1 = f_2 = \mathbb{I}_{x=1}$. Then the $f$-divergences $D_{f_1}(\gamma_x\|p)$ and $D_{f_2}(\gamma_y\|q)$ become infinity if $p \neq \gamma_x$ or $q \neq \gamma_y$, and become zeros otherwise.

**Dual form of unbalanced EOT problem** (3) is formulated as follows

$$\sup_{(\phi,\psi)} \left\{ -\varepsilon \int_{\mathbb{R}^d} \int_{\mathbb{R}^d} \exp\{\frac{1}{\varepsilon}(\phi(x)+\psi(y)-\frac{\|x-y\|^2}{2})\}dxdy - \int_{\mathbb{R}^d} \overline{f}_1(-\phi(x))p(x)dx - \int_{\mathbb{R}^d} \overline{f}_2(-\psi(y))q(y)dy \right\}. \tag{4}$$

It is known that there exist two measurable functions $\phi^*$, $\psi^*$ delivering maximum to (4). They have the following connection with the solution of the primal unbalanced problem (3):

$$\gamma^*(x,y) = \exp\{\frac{\phi^*(x)}{\varepsilon}\}\exp\{-\frac{\|x-y\|^2}{2\varepsilon}\}\exp\{\frac{\psi^*(y)}{\varepsilon}\}. \tag{5}$$

*Remark.* For some of our results, we will need to restrict potentials $(\phi, \psi)$ in problem (4) to the space $\mathcal{C}_{2,b}(\mathbb{R}^d) \times \mathcal{C}_{2,b}(\mathbb{R}^d)$. Since established variants of dual forms [8] typically correspond to other functional spaces, we derive and theoretically justify a variant of the *dual problem* (4) in Appendix A.3. Note that it may hold that optimal potentials $\psi^*, \phi^* \notin \mathcal{C}_{2,b}(\mathbb{R}^D)$, i.e., the supremum is not achieved within our considered spaces. This is not principle for our subsequent derivations.

**Computational UEOT setup.** Analytical solution for the *unbalanced* EOT problem is, in general, not known.[1] Moreover, in real-world setups where unbalanced EOT is applicable, the measures $p$, $q$ are typically not available explicitly but only through their empirical samples (datasets).

We assume that data measures $p, q \in \mathcal{P}_{2,ac}(\mathbb{R}^d)$ are unknown and accessible only by a limited number of i.i.d. empirical samples $\{x_0, ..., x_N\} \sim p$, $\{y_0, ..., y_M\} \sim q$. We aim to approximate the optimal UEOT plan $\gamma^*$ solving (3) between the entire measures $p$, $q$. The recovered plans should allow the out-of-sample estimation, i.e., generation of samples from $\gamma^*(\cdot|x^{\text{new}})$ where $x^{\text{new}}$ is a new test point (not necessarily present in the train data). Optionally, one may require the ability to sample from $\gamma_x^*$.

---

[1]Analytical solutions are known only for some specific cases. For example, [33] consider Gaussian measures and UEOT problem with $D_{\text{KL}}$ divergence instead of the differential entropy. This setup differs from ours.

| Solver | Problem | Principles | What recovers? | Limitations |
|---|---|---|---|---|
| [70] | UOT | Solves $c$-transform based semi-dual max-min reformulation of UOT using neural nets | Scaling factor $\gamma^*(x)/p(x)$ and stochastic OT map $T^*(x,z)$ | Complex max-min objective; 3 neural networks |
| [45] | Custom UOT | Regression on top of discrete EOT between re-balanced measures combined with ICNN-based solver [47] | Scaling factors and OT maps between re-scaled measures | Heuristically uses minibatch OT approximations |
| [9] | UOT | Solves semi-dual max-min reformulation of UOT using neural nets | Stochastic UOT map $T^*(x,z)$ | Complex max-min objective; 2 neural networks |
| [16] | UEOT | Flow Matching on top of discrete UEOT using neural nets | Parametrized vector field $(v_{t,\theta})_{t\in[0,1]}$ to transport the mass | Heuristically uses minibatch OT approximations |
| [36] | UEOT | Conditional Flow Matching on top of discrete EOT between re-balanced measures using neural nets | Scaling factors and parametrized conditional vector field $(v_{t,\theta})_{t\in[0,1]}$ to transport the mass between re-scaled measures | Heuristically uses minibatch OT approximations |
| U-LightOT (**ours**) | UEOT | Solves non-minimax reformulation of dual UEOT using Gaussian Mixtures | Density of UEOT plan $\gamma^*$ together with light procedure to sample $x \sim \gamma_x^*(\cdot)$ and $y \sim \gamma_y^*(\cdot\|x)$ | Restricted to Gaussian Mixture parametrization |

Table 1: Comparison of the principles of existing UOT/UEOT solvers and **our** proposed light solver.

The described setup is typically called the *continuous EOT* and should not be twisted up with the *discrete EOT* [53, 10]. There the aim is to recover the (unbalanced) EOT plan between the empirical measures $\hat{p}=\frac{1}{N}\sum_{i=1}^{N}\delta_{x_i}$, $\hat{q}=\frac{1}{M}\sum_{j=1}^{M}\delta_{y_j}$ and out-of-sample estimations are typically not needed.

## 3   Related Work

Nowadays, the sphere of continuous OT/EOT solvers is actively developing. Some of the early works related to this topic utilize OT cost as the loss function [27, 25, 2]. These approaches are not relevant to us as they do not learn OT/EOT maps (or plans). We refer to [39] for a detailed review.

At the same time, there exist a large amount of works within the discrete OT/EOT setup [10, 15, 69, 51], see [53] for a survey. We again emphasize that solvers of this type are not relevant to us as they construct discrete matching between the given (train) samples and typically do not provide a generalization to the new unseen (test) data. Only recently ML community started developing out-of-sample estimation procedures based on discrete/batched OT. For example, [19, 56, 32, 14, 48, 57] mostly develop such estimators using the barycentric projections of the discrete EOT plans. Though these procedures have nice theoretical properties, their scalability remains unclear.

**Balanced OT/EOT solvers.** There exists a vast amount of neural solvers for continuous OT problem. Most of them learn the OT maps (or plans) via solving saddle point optimization problems [3, 18, 37, 22, 60, 50]. Though the recent works [28, 61, 11, 41, 30] tackle the EOT problem (2), they consider its balanced version. Hence they are not relevant to us. Among these works, only [41, 30] evade non-trivial training/inference procedures and are ideologically the closest to ours. The difference between them consists of the particular loss function used. In fact, **our paper** proposes the solver which subsumes these solvers and generalizes them for the unbalanced case. The derivation of our solver is non-trivial and requires solid mathematical apparatus, see §4.

**Unbalanced OT/EOT solvers.** A vast majority of early works in this field tackle the discrete UOT/UEOT setup [6, 20, 54] but the principles behind their construction are not easy to generalize to the continuous setting. Thus, many of the recent papers that tackle the continuous unbalanced OT/EOT setup employ discrete solutions in the construction of their solvers. For example, [45] regress neural network on top of scaling factors obtained using the discrete UEOT while simultaneously learning the continuous OT maps using an ICNN method [47]. In [16] and [36], the authors implement Flow Matching [44, FM] and conditional FM on top of the discrete UEOT plans, respectively. The algorithm of the latter consists of regressing neural networks on top of scaling factors and simultaneously learning a conditional vector field to transport the mass between re-balanced measures. Despite the promising practical performance of these solvers, it is still unclear to what extent such approximations of UEOT plans are theoretically justified.

The recent papers [70, 9] are more related to our study as they do not rely on discrete OT approximations of the transport plan. However, they have non-trivial minimax optimization objectives solved using *complex* GAN-style procedures. Thus, these GANs often lean on heavy neural parametrization, may incur instabilities during training, and require careful hyperparameter selection [46].

For completeness, we also mention other papers which are only slightly relevant to us. For example, [23] considers incomplete OT which relaxes only one of the OT marginal constraints and is less general than the unbalanced OT. Other works [12, 4] incorporate unbalanced OT into the training objective of GANs aimed at generating samples from noise.

In contrast to the listed works, our paper proposes a *theoretically justified and lightweight* solver to the UEOT problem, see Table 1 for the detailed comparison of solvers.

# 4 Light Unbalanced OT Solver

In this section, we derive the optimization objective (§4.1) of our U-LightOT solver, present practical aspects of training and inference procedures (§4.2) and derive the generalization bounds for our solver (§4.4). We provide the *proofs of all theoretical results* in Appendix A.

## 4.1 Derivation of the Optimization Objective

Following the learning setup described above, we aim to get a parametric approximation $\gamma_{\theta,w}$ of the UEOT plan $\gamma^*$. Here $\theta, \omega$ are the model parameters to learn, and it will be clear later why we split them into two groups. To recover $\gamma_{\theta,\omega} \approx \gamma^*$, our aim is to learn $\theta, \omega$ by directly minimizing the $D_{KL}$ divergence between $\gamma_{\theta,w}$ and $\gamma^*$:

$$D_{KL}\left(\gamma^* \| \gamma_{\theta,w}\right) \to \min_{(\theta,w)}. \tag{6}$$

The main difficulty of this optimizing objective (6) is obvious: the UEOT plan $\gamma^*$ is *unknown*. Fortunately, below we show that one still can optimize (6) without knowing $\gamma^*$.

Recall that the optimal UEOT plan $\gamma^*$ has the form (5). We first make some changes of the variables. Specifically, we define $v^*(y) \stackrel{\text{def}}{=} \exp\{\frac{2\psi^*(y)-\|y\|^2}{2\varepsilon}\}$. Formula (5) now reads as

$$\gamma^*(x,y) = \exp\{\frac{2\phi^*(x)-\|x\|^2}{2\varepsilon}\}\exp\{\frac{\langle x,y\rangle}{\varepsilon}\}v^*(y) \Longrightarrow \gamma^*(y|x) \propto \exp\{\frac{\langle x,y\rangle}{\varepsilon}\}v^*(y). \tag{7}$$

Since the conditional plan has the unit mass, we may write

$$\gamma^*(y|x) = \exp\{\frac{\langle x,y\rangle}{\varepsilon}\}\frac{v^*(y)}{c_{v^*}(x)} \tag{8}$$

where $c_{v^*}(x) \stackrel{\text{def}}{=} \int_{\mathbb{R}^d} \exp\{\frac{\langle x,y\rangle}{\varepsilon}\}v^*(y)dy$ is the normalization costant ensuring that $\int_{\mathbb{R}^d} \gamma^*(y|x)dy = 1$.

Consider the decomposition $\gamma^*(x,y) = \gamma_x^*(x)\gamma^*(y|x)$. It shows that to obtain parametrization for the entire plan $\gamma^*(x,y)$, it is sufficient to consider parametrizations for its left marginal $\gamma_x^*$ and the conditional measure $\gamma^*(y|x)$. Meanwhile, equation (8) shows that *conditional measures $\gamma^*(\cdot|x)$ are entirely described by the variable $v^*$*. We use these observations to parametrize $\gamma_{\theta,w}$. We set

$$\gamma_{\theta,w}(x,y) \stackrel{\text{def}}{=} u_\omega(x)\gamma_\theta(y|x) = u_\omega(x)\frac{\exp\{\langle x,y\rangle/\varepsilon\}v_\theta(y)}{c_\theta(x)}, \tag{9}$$

where $u_\omega$ and $v_\theta$ parametrize marginal measure $\gamma_x^*$ and the variable $v^*$, respectively. In turn, the constant $c_\theta(x) \stackrel{\text{def}}{=} \int_{\mathbb{R}^d} \exp\{\frac{\langle x,y\rangle}{\varepsilon}\}v_\theta(y)dy$ is the parametrization of $c_{v^*}(x)$. Next, we demonstrate our **main result** which shows that the optimization of (6) can be done *without* the access to $\gamma^*$.

**Theorem 4.1** (Tractable form of $D_{KL}$ minimization). *Assume that $\gamma^*$ is parametrized using* (9). *Then the following bound holds:* $\varepsilon D_{KL}\left(\gamma^*\|\gamma_{\theta,w}\right) \leq \mathcal{L}(\theta,w) - \mathcal{L}^*$, *where*

$$\mathcal{L}(\theta,w) \stackrel{def}{=} \int_{\mathbb{R}^d} \overline{f}_1\left(-\varepsilon\log\frac{u_\omega(x)}{c_\theta(x)} - \frac{\|x\|^2}{2}\right)p(x)dx + \int_{\mathbb{R}^d} \overline{f}_2\left(-\varepsilon\log v_\theta(y) - \frac{\|y\|^2}{2}\right)q(y)dy + \varepsilon\|u_\omega\|_1, \tag{10}$$

*and constant $(-\mathcal{L}^*)$ is the optimal value of the dual form* (4). *The bound is **tight** in the sence that it turns to $0 = 0$ when $v_\theta(y) = \exp\{2\psi^*(y)-\|y\|^2/2\varepsilon\}$ and $u_\omega(x) = c_\theta(x)\exp\{2\phi^*(x)-\|x\|^2/2\varepsilon\}$.*

In fact, (10) is the dual form (4) but with potentials $(\phi,\psi)$ expressed through $u_\omega, v_\theta$ (and $c_\theta$):

$$\phi(x) \leftarrow \phi_{\theta,\omega}(x) = \varepsilon\log\frac{u_\omega(x)}{c_\theta(x)} + \frac{\|x\|^2}{2}, \quad \psi(y) \leftarrow \psi_\theta(y) = \varepsilon\log v_\theta(y) + \frac{\|y\|^2}{2}.$$

Our result can be interpreted as the bound on the quality of approximate solution $\gamma_{\theta,\omega}$ (3) recovered from the approximate solution to the dual problem (4). It can be directly proved using the original $(\phi,\psi)$ notation of the dual problem, but we use $(u_\omega, v_\theta)$ instead as with this change of variables the form of the recovered plan $\gamma_{\theta,\omega}$ is more interpretable ($u_w$ defines the first marginal, $v_\theta$ – conditionals).

Instead of optimizing (6) to get $\gamma_{\theta,\omega}$, we may optimize the upper bound (10) which is more tractable. Indeed, (10) is a sum of the expectations w.r.t. the probability measures $p, q$. We can obtain Monte-Carlo estimation of (10) from random samples and optimize it with stochastic gradient descent procedure w.r.t. $(\theta,\omega)$. The main **challenge** here is the computation of the variable $c_\theta$ and term $\|u_\omega\|_1$. Below we propose a smart parametrization by which both variables can be derived analytically.

## 4.2 Parametrization and the Optimization Procedure

**Parametrization.** Recall that $u_\omega$ parametrizes the density of the marginal $\gamma_x^*$ which is unnormalized. Setting $x = 0$ in equation (7), we get $\gamma^*(y|x=0) \propto v^*(y)$ which means that $v^*$ also corresponds to an unnormalized density of a measure. These motivate us to use the unnormalized Gaussian mixture parametrization for the potential $v_\theta(y)$ and measure $u_\omega(x)$:

$$v_\theta(y) = \sum_{k=1}^{K} \alpha_k \mathcal{N}(y|r_k, \varepsilon S_k); \quad u_\omega(x) = \sum_{l=1}^{L} \beta_l \mathcal{N}(x|\mu_l, \varepsilon \Sigma_l). \tag{11}$$

Here $\theta \stackrel{\text{def}}{=} \{\alpha_k, r_k, S_k\}_{k=1}^{K}$, $w \stackrel{\text{def}}{=} \{\beta_l, \mu_l, \Sigma_l\}_{l=1}^{L}$ with $\alpha_k, \beta_l \geq 0$, $r_k, \mu_l \in \mathbb{R}^d$ and $0 \prec S_k, \Sigma_l \in \mathbb{R}^{d \times d}$. The covariance matrices are scaled by $\varepsilon$ just for convenience.

For this type of parametrization, it holds that $\|u_\omega\|_1 = \sum_{l=1}^{L} \beta_l$. Moreover, there exist closed-from expressions for the normalization constant $c_\theta(x)$ and conditional plan $\gamma_\theta(y|x)$, see [41, Proposition 3.2]. Specifically, define $r_k(x) \stackrel{\text{def}}{=} r_k + S_k x$ and $\widetilde{\alpha}_k(x) \stackrel{\text{def}}{=} \alpha_k \exp\{\frac{x^T S_k x + 2 r_k^T x}{2\varepsilon}\}$. It holds that

$$c_\theta(x) = \sum_{k=1}^{K} \widetilde{\alpha}_k(x); \quad \gamma_\theta(y|x) = \frac{1}{c_\theta(x)} \sum_{k=1}^{K} \widetilde{\alpha}_k(x) \mathcal{N}(y|r_k(x), \varepsilon S_k). \tag{12}$$

Using this result and (11), we get the expression for $\gamma_{\theta, w}$:

$$\gamma_{\theta, w}(x, y) = u_\omega(x) \cdot \gamma_\theta(y|x) = \underbrace{\sum_{l=1}^{L} \beta_l \mathcal{N}(x|\mu_l, \varepsilon \Sigma_l)}_{u_\omega(x)} \cdot \underbrace{\frac{\sum_{k=1}^{K} \widetilde{\alpha}_k(x) \mathcal{N}(y|r_k(x), \varepsilon S_k)}{\sum_{k=1}^{K} \widetilde{\alpha}_k(x)}}_{\gamma_\theta(y|x)}. \tag{13}$$

**Training.** We recall that the measures $p, q$ are accessible only by a number of empirical samples (see the learning setup in §2). Thus, given samples $\{x_1, ..., x_N\}$ and $\{y_1, ..., y_M\}$, we optimize the empirical analog of (10):

$$\widehat{\mathcal{L}}(\theta, w) \stackrel{\text{def}}{=} \frac{1}{N} \sum_{i=1}^{N} \overline{f}_1(-\varepsilon \log \frac{u_\omega(x_i)}{c_\theta(x_i)} - \frac{\|x_i\|^2}{2}) + \frac{1}{M} \sum_{j=1}^{M} \overline{f}_2(-\varepsilon \log v_\theta(y_j) - \frac{\|y_j\|^2}{2}) + \varepsilon \|u_\omega\|_1 \tag{14}$$

using minibatch gradient descent procedure w.r.t. parameters $(\theta, w)$. In the parametrization of $v_\theta$ and $u_\omega$ (11), we utilize the diagonal matrices $S_k$, $\Sigma_l$. This allows decreasing the number of learnable parameters in $\theta$ and speeding up the computation of inverse matrices $S_k^{-1}$, $\Sigma_l^{-1}$.

**Inference.** Sampling from the conditional and marginal measures $\gamma_\theta(y|x) \approx \gamma^*(y|x)$, $u_\omega \approx \gamma_x^*$ is easy and lightweight since they are explicitly parametrized as Gaussian mixtures, see (12), (11).

## 4.3 Connection to Related Prior Works

The idea of using Gaussian Mixture parametrization for dual potentials in EOT-related tasks first appeared in the EOT/SB benchmark [29]. There it was used to obtain the benchmark pairs of probability measures with the known EOT solution between them. In [41], the authors utilized this type of parametrization to obtain a light solver (**LightSB**) for the **balanced** EOT.

Our solver for **unbalanced** EOT (10) subsumes their solver for balanced EOT as well as one problem subsumes the other for the special case of divergences, see the remark in §2. Let $f_1$, $f_2$ be convex indicators of $\{1\}$. Then $\overline{f_1}(t) = t$ and $\overline{f_2}(t) = t$ and objective (10) becomes

$$\mathcal{L}(\theta, w) = \int_{\mathbb{R}^d} (-\varepsilon \log \frac{u_\omega(x)}{c_\theta(x)} - \frac{\|x\|^2}{2}) p(x) dx + \int_{\mathbb{R}^d} (-\varepsilon \log v_\theta(y) - \frac{\|y\|^2}{2}) q(y) dy + \varepsilon \|u_\omega\|_1 =$$

$$-\varepsilon \Big( \int_{\mathbb{R}^d} \log \frac{u_\omega(x)}{c_\theta(x)} p(x) dx + \int_{\mathbb{R}^d} \log v_\theta(y) q(y) dy - \|u_\omega\|_1 \Big) - \underbrace{\int_{\mathbb{R}^d} \frac{\|x\|^2}{2} p(x) dx - \int_{\mathbb{R}^d} \frac{\|y\|^2}{2} q(y) dy}_{\stackrel{\text{def}}{=} \text{Const}(p,q)} =$$

$$\varepsilon \Big( \int_{\mathbb{R}^d} \log c_\theta(x) p(x) dx - \int_{\mathbb{R}^d} \log v_\theta(y) q(y) dy \Big) - \tag{15}$$

$$\varepsilon \int_{\mathbb{R}^d} \log u_\omega(x) p(x) dx + \varepsilon \|u_\omega\|_1 - \text{Const}(p, q). \tag{16}$$

Here line (15) depends exclusively on $\theta$ and exactly coincides with the LightSB's objective, see [41, Proposition 8]. At the same time, line (16) depends only on $\omega$, and its minimum is attained when $u_\omega = p$. Thus, this part is not actually needed in the balanced case, see [41, Appendix C].

## 4.4 Generalization Bounds and Universal Approximation Property

Theoretically, to recover the UEOT plan $\gamma^*$, one needs to solve the problem $\mathcal{L}(\theta, \omega) \to \min_{\theta, \omega}$ as stated in our Theorem 4.1. In practice, the measures $p$ and $q$ are accessible via empirical samples $X \overset{\text{def}}{=} \{x_1, ..., x_N\}$ and $Y \overset{\text{def}}{=} \{y_1, ..., y_M\}$, thus, one needs to optimize the empirical counterpart $\widehat{\mathcal{L}}(\theta, \omega)$ of $\mathcal{L}(\theta, \omega)$, see (14). Besides, the available potentials $u_\omega$, $v_\theta$ over which one optimizes the objective come from the restricted class of functions. Specifically, we consider unnormalized Gaussian mixtures $u_\omega$, $v_\theta$ with $K$ and $L$ components respectively. Thus, one may naturally wonder: **how close is the recovered plan $\gamma_{\widehat{\theta}, \widehat{\omega}}$ to the UEOT plan** $\gamma^*$ given that $(\widehat{\theta}, \widehat{\omega}) = \arg\min_{\theta, \omega} \widehat{\mathcal{L}}(\theta, \omega)$?

To address this question, we study the *generalization error* $\mathbb{E}D_{\text{KL}}\left(\gamma^* \| \gamma_{\widehat{\theta}, \widehat{\omega}}\right)$, i.e., the expectation of $D_{\text{KL}}$ between $\gamma^*$ and $\gamma_{\widehat{\theta}, \widehat{\omega}}$ taken w.r.t. the random realization of the train datasets $X \sim p$, $Y \sim q$.

Let $(\overline{\theta}, \overline{\omega}) = \arg\min_{(\theta, \omega) \in \Theta \times \Omega} \mathcal{L}(\theta, \omega)$ denote the best approximators of $\mathcal{L}(\theta, \omega)$ in the admissible class. From Theorem 4.1 it follows that that $\mathbb{E}D_{\text{KL}}\left(\gamma^* \| \gamma_{\widehat{\theta}, \widehat{\omega}}\right)$ can be upper bounded using $\mathbb{E}(\mathcal{L}(\widehat{\theta}, \widehat{\omega}) - \mathcal{L}^*)$. The latter can be decomposed into the estimation and approximation errors:

$$\mathbb{E}(\mathcal{L}(\widehat{\theta}, \widehat{\omega}) - \mathcal{L}^*) = \mathbb{E}[\mathcal{L}(\widehat{\theta}, \widehat{\omega}) - \mathcal{L}(\overline{\theta}, \overline{\omega})] + \mathbb{E}[\mathcal{L}(\overline{\theta}, \overline{\omega}) - \mathcal{L}^*] = \underbrace{\mathbb{E}[\mathcal{L}(\widehat{\theta}, \widehat{\omega}) - \mathcal{L}(\overline{\theta}, \overline{\omega})]}_{\text{Estimation error}} + \underbrace{[\mathcal{L}(\overline{\theta}, \overline{\omega}) - \mathcal{L}^*]}_{\text{Approximation error}} . \quad (17)$$

We establish the quantitative bound for the estimation error in the proposition below.

**Proposition 4.2** (Bound for the estimation error). *Let $p, q$ be compactly supported and assume that $\overline{f}_1$, $\overline{f}_2$ are Lipshitz. Assume that the considered parametric classes $\Theta$, $\Omega$ ($\ni (\theta, \omega)$) consist of unnormalized Gaussian mixtures with $K$ and $L$ components respectively with bounded means $\|r_k\|, \|\mu_l\| \leq R$ (for some $R > 0$), covariances $sI \preceq S_k, \Sigma_l \preceq SI$ (for some $0 < s \leq S$) and weights $a \leq \alpha_k, \beta_l \leq A$ (for some $0 < a \leq A$). Then the following holds:*

$$\mathbb{E}\left[\mathcal{L}(\widehat{\theta}, \widehat{\omega}) - \mathcal{L}(\overline{\theta}, \overline{\omega})\right] \leq O(\frac{1}{\sqrt{N}}) + O(\frac{1}{\sqrt{M}}),$$

*where $O(\cdot)$ hides the constants depending only on $K, L, R, s, S, a, A, p, q, \varepsilon$ but not on sizes $M, N$.*

This proposition allows us to conclude that the estimation error converges to zero when $N$ and $M$ tend to infinity at the usual parametric rate. It remains to clarify the question: *can we make the approximation error arbitrarily small*? We answer this question positively in our Theorem below.

**Theorem 4.3** (Gaussian mixture parameterization for the variables provides the universal approximation of UEOT plans). *Let $p$ and $q$ be compactly supported and assume that $\overline{f}_1$, $\overline{f}_2$ are Lipshitz. Then for all $\delta > 0$ there exist Gaussian mixtures $v_\theta$, $u_\omega$ (11) with **scalar** covariances $S_k = \lambda_k I_d \succ 0$, $\Sigma_l = \zeta_l I_d \succ 0$ of their components that satisfy $D_{KL}\left(\gamma^* \| \gamma_{\theta, \omega}\right) \leq \varepsilon^{-1}(\mathcal{L}(\theta, \omega) - \mathcal{L}^*) < \delta$.*

**In summary**, results of this section show that one can make the generalization error *arbitrarily small* given a sufficiently large amount of samples and components in the Gaussian parametrization. It means that theoretically our solver can recover the UEOT plan arbitrarily well.

## 5 Experiments

In this section, we test our U-LightOT solver on several setups from the related works. The code is written using `PyTorch` framework and is publicly available at

$$\texttt{https://github.com/milenagazdieva/LightUnbalancedOptimalTransport}.$$

The experiments are issued in the convenient form of `*.ipynb` notebooks. Each experiment requires several minutes of training on CPU with 4 cores. Technical *training details* are given in Appendix B.

### 5.1 Example with the Mixture of Gaussians

We provide an illustrative *'Gaussians Mixture'* example in 2D to demonstrate the ability of our solver to deal with the imbalance of classes in the source and target measures. We follow the experimental

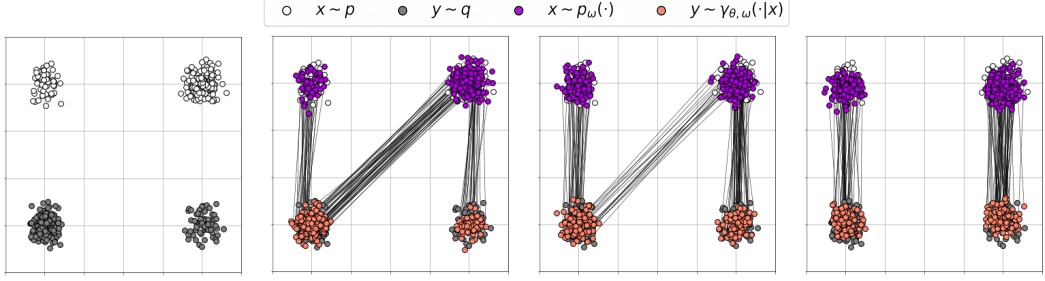

| (a) Input, target measures | (b) U-LightOT, $\tau = 10^2$ | (c) U-LightOT, $\tau = 10^1$ | (d) U-LightOT, $\tau = 10^0$ |

Figure 2: Conditional plans $\gamma_{\theta,\omega}(y|x)$ learned by our solver in *Gaussians Mixture* experiment with unbalancedness parameter $\tau \in [10^0, 10^1, 10^2]$. Here $p_\omega$ denotes the normalized first marginal $u_w$, i.e., $p_\omega = u_\omega / \|u_\omega\|_1$.

setup proposed in [16, Figure 2] and define the probability measures $p$, $q$ as follows (Fig. 2a):

$$p(x) = \frac{1}{4}\mathcal{N}(x|(-3,3), 0.1 \cdot I_2) + \frac{3}{4}\mathcal{N}(x|(1,3), 0.1 \cdot I_2),$$

$$q(y) = \frac{3}{4}\mathcal{N}(y|(-3,0), 0.1 \cdot I_2) + \frac{1}{4}\mathcal{N}(y|(1,0), 0.1 \cdot I_2).$$

We test our U-LightOT solver with *scaled* $D_{\text{KL}}$ divergences, i.e., $f_1(t), f_2(t)$ are defined by $\tau \cdot f_{\text{KL}}(t) = \tau(t\log(t) - t + 1)$ where $\tau > 0$. We provide the learned plans for $\tau \in [1, 10^1, 10^2]$. The results in Fig. 2 show that parameter $\tau$ can be used to control the level of unbalancedness of the learned plans. For $\tau = 1$, our U-LightOT solver truly learns the UEOT plans, see Fig. 2d. When $\tau$ increases, the solver fails to transport the mass from the input Gaussians to the closest target ones. Actually, for $\tau = 10^2$, our solutions are similar to the solutions of [41, LightSB] solver which approximates balanced EOT plans between the measures. Hereafter, we treat $\tau$ as the *unbalancedness* parameter. In Appendix C, we test the performance of our solver with $D_{\chi^2}$ divergence.

*Remark.* Here we conduct all experiments with the entropy regularization parameter $\varepsilon = 0.05$. The parameter $\varepsilon$ is responsible for the stochasticity of the learned transport $\gamma_\theta(\cdot|x)$. Since we are mostly interested in the correct transport of the mass (controlled by $f_1, f_2$) rather than the stochasticity, we do not pay much attention to $\varepsilon$ throughout the paper.

## 5.2 Unpaired Image-to-Image Translation

Another popular testbed which is usually considered in OT/EOT papers is the unpaired image-to-image translation [71] task. Since our solver uses the parametrization based on Gaussian mixture, it may be hard to apply U-LightOT for learning translation directly in the image space. Fortunately, nowadays it is common to use autoencoders [59] for more efficient translation. We follow the setup of [41, Section 5.4] and use pre-trained ALAE autoencoder [55] for $1024 \times 1024$ FFHQ dataset [34] of human faces. We consider different subsets of FFHQ dataset (*Adult*, *Young*, *Man*, *Woman*) and all variants of translations between them: *Adult↔Young* and *Woman↔Man*.

The main challenge of the described translations is the **imbalance** of classes in the images from source and target subsets, see Table 2. Let us consider in more detail *Adult→Young* translation. In the FFHQ dataset, the amount of *adult men* significantly outnumbers the *adult women*, while the amount of *young men* is smaller than that of *young women*. Thus, balanced OT/EOT solvers are expected to translate some of *adult man* representatives to *young woman* ones. At the same time, solvers based on unbalanced transport are supposed to alleviate this issue.

| **Class** | *Man* | *Woman* |
|---|---|---|
| *Young* | $15K$ | $23K$ |
| *Adult* | $7K$ | $3.5K$ |

Table 2: Number of *train* FFHQ images for each subset.

**Baselines.** We perform a comparison with the recent procedure [16, UOT-FM] which considers roughly the same setup and demonstrates good performance. This method interpolates the results of unbalanced discrete OT to the continuous setup using the flow matching [44]. For completeness, we include the comparison with LightSB and balanced optimal transport-based flow matching (OT-FM) [44] to demonstrate the issues of the balanced solvers. We also consider neural network based solvers relying on the adversarial learning such as UOT-SD [9] and UOT-GAN[70].

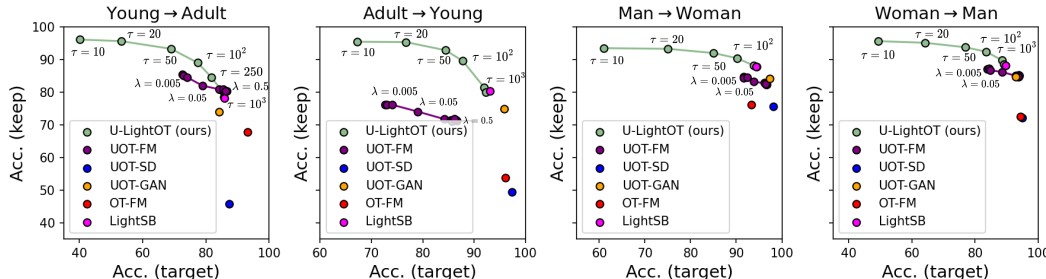

Figure 3: Visualization of pairs of accuracies (*keep-target*) for our U-LightOT solver and other OT/EOT methods in the image translation experiment. The values of unbalancedness parameters for our U-LightOT solver ($\tau$) and [16, UOT-FM] ($\lambda = reg\_m$) are specified directly on the plots.

| Experiment | [16, UOT-FM] | [9, UOT-SD] | [70, UOT-GAN] | U-LightOT (**ours**) |
|---|---|---|---|---|
| *Time (sec)* | 03:21 | 18:11 | 16:30 | **02:38** |

Table 3: Comparison of wall-clock running times of unbalanced OT/EOT solvers in *Young→Adult* translation. The best results are in **bold**, second best are underlined.

**Metrics.** We aim to assess the ability of the solvers to perform the translation of latent codes keeping the class of the input images unchanged, e.g., keeping the gender in *Adult→Young* translation. Thus, we train a 99% MLP classifier for gender using the latent codes of images. We use it to compute the accuracy of preserving the gender during the translation. We denote this number by *accuracy (keep)*.

Meanwhile, it is also important to ensure that the generated images belong to the distribution of the target images rather than the source ones, e.g., belong to the distribution of *young* people in our example. To monitor this property, we use another 99% MLP classifier to identify whether each of the generated images belong to the target domain or to the source one. Then we calculate the fraction of the generated images belonging to the target domain which we denote as *accuracy (target)*.

**Results.** Unbalanced OT/EOT approaches are equipped with some kind of unbalancedness parameter (like $\tau$ in our solver) which influences the methods' results: with the increase of unbalancedness, accuracy of keeping the class increases but the accuracy of mapping to the target decreases. The latter is because of the relaxation of the marginal constraints in UEOT. For a fair comparison, we aim to compute the trade-offs between the keep/target accuracies for different values of the unbalancedness. We do this for our solver and UOT-FM. We found that GAN-based unbalanced approaches show unstable behaviour, so we report UOT-SD and UOT-GAN's results only for one parameter value.

For convenience, we visualize the accuracies' pairs for our solver and its competitors in Fig. 3. The evaluation shows that our U-LightOT method can effectively solve translation tasks in high dimensions ($d = 512$) and outperforms its alternatives in dealing with class imbalance issues. Namely, we see that our solver allows for achieving the best accuracy of keeping the attributes of the input images than other methods while it provides good accuracy of mapping to the target class. While balanced methods and some of the unbalanced ones provide really high accuracy of mapping to the target, their corresponding accuracies of keeping the attributes are worse than ours meaning that we are better in dealing with class imbalance issues. As expected, for large parameter $\tau$, the results of our U-LightOT solver coincide with those for LightSB which is a balanced solver. Meanwhile, our solver has the lowest wall-clock running time among the existing unbalanced solvers, see Table 3 for comparison. We demonstrate qualitative results of our solver and baselines in Fig. 4. The choice of unbalancedness parameter for visualization of our method and UOT-FM is detailed in Appendix B.

We present the results of the *quantitative comparison* in the form of tables in Appendix D.1. In Appendix C, we perform the *ablation study* of our solver focusing on the selection of parameters $\tau$, $\varepsilon$ and number of Gaussian mixtures' components.

## 6 Discussion

**Potential impact.** Our light and unbalanced solver has a lot of advantages in comparison with the other existing UEOT solvers. First, it does not require complex max-min optimization. Second, it provides the closed form of the conditional measures $\gamma_\theta(y|x) \approx \gamma^*(y|x)$ of the UEOT plan.

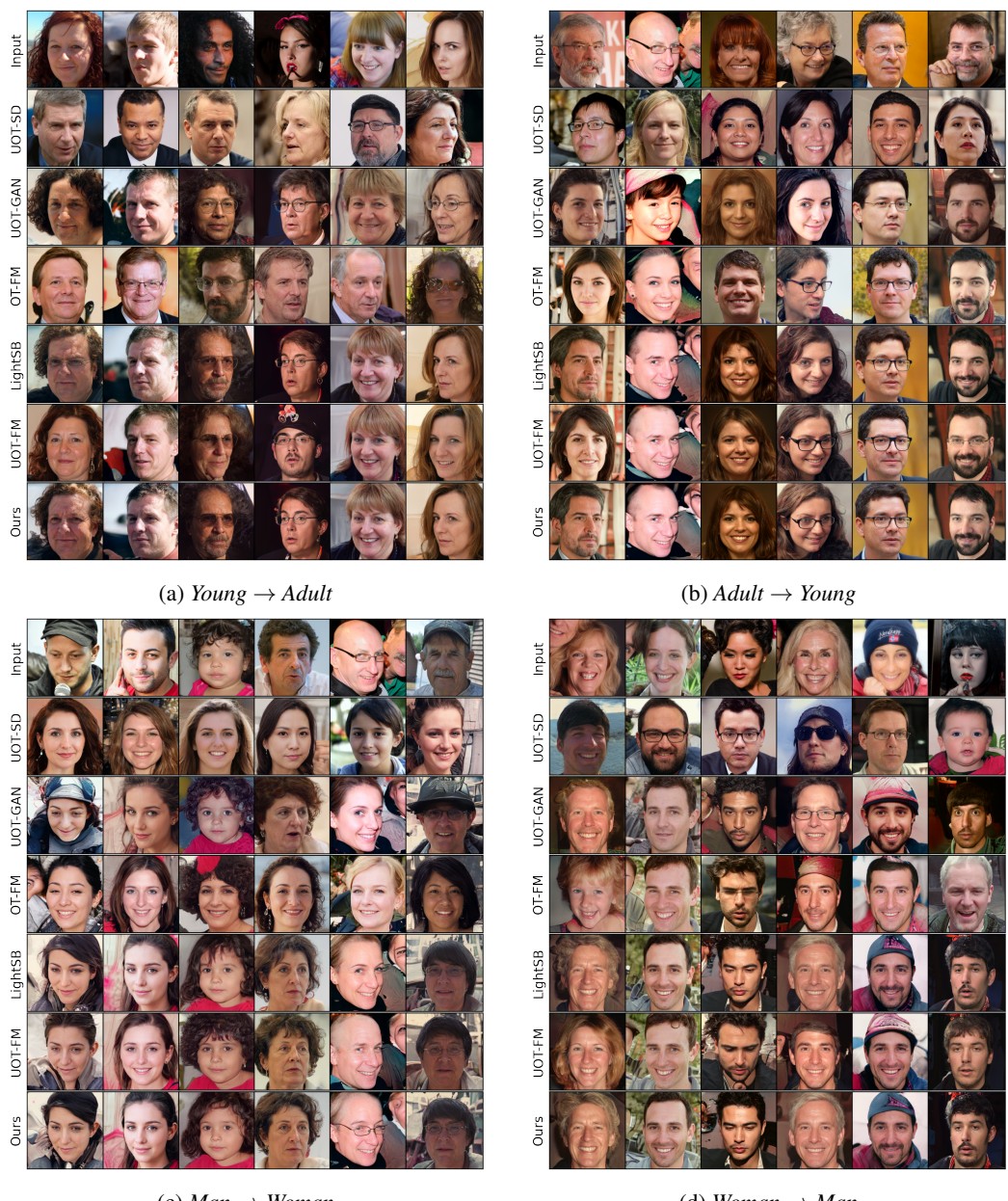

(a) *Young → Adult*

(b) *Adult → Young*

(c) *Man → Woman*

(d) *Woman → Man*

Figure 4: Unpaired translation with LightSB, OT-FM, UOT-FM and our U-LightOT solvers applied in the latent space of ALAE [55] for FFHQ images [34] (1024×1024).

Moreover, it allows for sampling both from the conditional measure $\gamma_\theta(y|x)$ and marginal measure $u_\omega(x) \approx \gamma_x^*(x)$. Besides, the decisive superiority of our lightweight and unbalanced solver is its simplicity and convenience of use. Indeed, it has a straightforward and non-minimax optimization objective and avoids heavy neural parametrization. As a result, our lightweight and unbalanced solver converges in minutes on CPU. We expect that these advantages could boost the usage of our solver as a standard and easy baseline for UEOT task with applications in different spheres.

The limitations and broader impact of our solver are discussed in Appendix E.

ACKNOWLEDGEMENTS. The work of Skoltech was supported by the Analytical center under the RF Government (subsidy agreement 000000D730321P5Q0002, Grant No. 70-2021-00145 02.11.2021). We thank Kirill Sokolov, Mikhail Persiianov and Petr Mokrov for providing valuable feedback and suggestions for improving the proofs and clarity of our paper.

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

# A  Proofs

## A.1  Proof of Theorem 4.1

*Proof.* For convenience, we split this proof into several steps.

*Step 1.* To begin with, we recall that the primal UEOT problem (3) attains a minimizer $\gamma^*$. Then by substituting $\gamma^*$ directly in (3), we get

$$(3) = \int_{\mathbb{R}^d}\int_{\mathbb{R}^d}\frac{\|x-y\|^2}{2}\gamma^*(x,y)dxdy - \varepsilon H(\gamma^*) + D_{f_1}\left(\gamma^*_x\|p\right) + D_{f_2}\left(\gamma^*_y\|q\right) =$$

$$\int_{\mathbb{R}^d}\int_{\mathbb{R}^d}\frac{\|x-y\|^2}{2}\gamma^*(x,y)dxdy + \varepsilon\int_{\mathbb{R}^d}\int_{\mathbb{R}^d}\gamma^*(x,y)\log\gamma^*(x,y)dxdy - \varepsilon\|\gamma^*\|_1 +$$

$$\int_{\mathbb{R}^d}f_1\left(\frac{\gamma^*_x(x)}{p(x)}\right)p(x)dx + \int_{\mathbb{R}^d}f_2\left(\frac{\gamma^*_y(y)}{q(y)}\right)q(y)dy = -\varepsilon\|\gamma^*\|_1 +$$

$$\int_{\mathbb{R}^d}\int_{\mathbb{R}^d}\frac{\|x-y\|^2}{2}\gamma^*(x,y)dxdy + \varepsilon\int_{\mathbb{R}^d}\gamma^*(x,y)\cdot\underbrace{\varepsilon^{-1}\left(\phi^*(x)+\psi^*(y)-\frac{\|x-y\|^2}{2}\right)}_{\log\gamma^*(x,y)=}dxdy + \quad(18)$$

$$\int_{\mathbb{R}^d}f_1\left(\frac{\gamma^*_x(x)}{p(x)}\right)p(x)dx + \int_{\mathbb{R}^d}f_2\left(\frac{\gamma^*_y(y)}{q(y)}\right)q(y)dy = -\varepsilon\|\gamma^*\|_1 +$$

$$\int_{\mathbb{R}^d}\gamma^*_x(x)\phi^*(x)dx + \int_{\mathbb{R}^d}\gamma^*_y(y)\psi^*(y)dy + \int_{\mathbb{R}^d}f_1\left(\frac{\gamma^*_x(x)}{p(x)}\right)p(x)dx + \int_{\mathbb{R}^d}f_2\left(\frac{\gamma^*_y(y)}{q(y)}\right)q(y)dy =$$

$$-\varepsilon\|\gamma^*\|_1 + \int_{\mathbb{R}^d}\left(f_1\left(\frac{\gamma^*_x(x)}{p(x)}\right)+\phi^*(x)\frac{\gamma^*_x(x)}{p(x)}\right)p(x)dx + \int_{\mathbb{R}^d}\left(f_2\left(\frac{\gamma^*_y(y)}{q(y)}\right)+\psi^*(y)\frac{\gamma^*_y(y)}{q(y)}\right)q(y)dy. \quad(19)$$

Here in line (18), we use equation (5) which establishes the connection between $\gamma^*$ and the optimal potentials $\phi^*$, $\psi^*$ delivering maximum to the dual UEOT problem (4). Substituting these optimal potentials to the dual problem, we get

$$(4) = -\varepsilon\underbrace{\int_{\mathbb{R}^d}\int_{\mathbb{R}^d}\exp\{\frac{1}{\varepsilon}(\phi^*(x)+\psi^*(y)-\frac{\|x-y\|^2}{2})\}dxdy}_{=\|\gamma^*\|_1} - \int_{\mathbb{R}^d}\overline{f}_1(-\phi^*(x))p(x)dx -$$

$$\int_{\mathbb{R}^d}\overline{f}_2(-\psi^*(y))q(y)dy = -\varepsilon\|\gamma^*\|_1 - \int_{\mathbb{R}^d}\overline{f}_1(-\phi^*(x))p(x)dx - \int_{\mathbb{R}^d}\overline{f}_2(-\psi^*(y))q(y)dy. \quad(20)$$

Note that (19) equals to (20) due to the equivalence between primal and dual UEOT problems, see §2. We will use this fact in further derivations.

For every function $f$, its convex conjugate

$$\overline{f}(t) = \sup_{u\in\mathbb{R}}\{ut - f(u)\} \Longrightarrow -\overline{f}(-t) = -\sup_{u\in\mathbb{R}}\{-ut - f(u)\} = \inf_{u\in\mathbb{R}}\{ut + f(u)\} \Longrightarrow$$

$$-\overline{f}(-t) \leq ut + f(u) \quad \forall u\in\mathbb{R}. \quad(21)$$

Then for arbitrary $x\in\mathbb{R}^d$, taking $f = f_1$, $u = \frac{\gamma^*_x(x)}{p(x)}$, $t = \phi^*(x)$, we get

$$-\overline{f_1}(-\phi^*(x)) \leq \frac{\gamma^*_x(x)}{p(x)}\phi^*(x) + f_1\left(\frac{\gamma^*_x(x)}{p(x)}\right) \Longrightarrow \quad(22)$$

$$-\int_{\mathbb{R}^d}\overline{f_1}(\phi^*(x))p(x)dx \leq \int_{\mathbb{R}^d}\left(\frac{\gamma^*_x(x)}{p(x)}\phi^*(x) + f_1\left(\frac{\gamma^*_x(x)}{p(x)}\right)\right)p(x)dx. \quad(23)$$

Similarly, for arbitrary $y\in\mathbb{R}^d$

$$-\int_{\mathbb{R}^d}\overline{f_2}(\psi^*(y))q(y)dy \leq \int_{\mathbb{R}^d}\left(\frac{\gamma^*_y(y)}{q(y)}\psi^*(y) + f_2\left(\frac{\gamma^*_y(y)}{q(y)}\right)\right)q(y)dy. \quad(24)$$

Since (19)=(20), inequalities (22), (23), (24) actually turn to equalities.

*Step 2.* Since we consider convex, lower semi-continuous functions $f$, it holds that $\overline{\overline{f}} = f$. Then, substituting $\overline{f}$ in formula (21), we derive

$$-f(-t) = -\overline{\overline{f}}(-t) = \inf_{u \in \mathbb{R}} \{ut + \overline{f}(u)\} \implies -f(-t) \leq ut + \overline{f}(u) \ \forall u \in \mathbb{R}. \tag{25}$$

Thus, for arbitrary $x \in \mathbb{R}^d$, taking $f = f_1$, $t = -\frac{\gamma_x^*(x)}{p(x)}$, we get that $-f_1\left(\frac{\gamma_x^*(x)}{p(x)}\right) \leq u\frac{\gamma_x^*(x)}{p(x)} + \overline{f_1}(-u) \ \forall u \in \mathbb{R}$. At the same time, from *step 1*, we know that $-f_1\left(\frac{\gamma_x^*(x)}{p(x)}\right) = \frac{\gamma_x^*(x)}{p(x)}\phi^*(x) + \overline{f_1}(-\phi^*(x))$, see (22). Then

$$\frac{\gamma_x^*(x)}{p(x)}\phi^*(x) + \overline{f_1}(-\phi^*(x)) \leq u\frac{\gamma_x^*(x)}{p(x)} + \overline{f_1}(-u) \ \forall u \in \mathbb{R} \implies$$

$$\frac{\gamma_x^*(x)}{p(x)}(\phi^*(x) - u) \leq (\overline{f_1}(-u) - \overline{f_1}(-\phi^*(x)) \ \forall u \in \mathbb{R}. \tag{26}$$

Similarly, for arbitrary $y \in \mathbb{R}^d$

$$\frac{\gamma_y^*(y)}{q(y)}(\psi^*(y) - u) \leq (\overline{f_2}(-u) - \overline{f_2}(-\psi^*(y)) \ \forall u \in \mathbb{R}. \tag{27}$$

*Step 3.* Now we are ready to prove the main result. We note that:

$$\mathrm{D_{KL}}\left(\gamma^* \| \gamma_{\theta,\omega}\right) = \int_{\mathbb{R}^d \times \mathbb{R}^d} \gamma^*(x,y) \log \gamma^*(x,y) dx dy -$$

$$\int_{\mathbb{R}^d \times \mathbb{R}^d} \gamma^*(x,y) \log \gamma_{\theta,\omega}(x,y) dx dy + \|\gamma_{\theta,\omega}\|_1 - \|\gamma^*\|_1. \tag{28}$$

While the ground-truth UEOT plan $\gamma^*(x,y)$ and the optimal dual variables $\phi^*(x)$, $\psi^*(y)$ are connected via equation (5), our parametrized plan $\gamma_{\theta,\omega}(x,y)$ can be expressed using $\phi_{\theta,\omega}(x)$, $\psi_\theta(y)$ as $\gamma_{\theta,\omega}(x,y) = \exp\{\varepsilon^{-1}(\phi_{\theta,\omega}(x) + \psi_\theta(y) - \|x-y\|^2/2)\}$, see (9). Then

$$(28) = \varepsilon^{-1} \int_{\mathbb{R}^d \times \mathbb{R}^d} \gamma^*(x,y)\big(\phi^*(x) + \psi^*(y) - \|x-y\|^2/2\big) dx dy -$$

$$\varepsilon^{-1} \int_{\mathbb{R}^d \times \mathbb{R}^d} \gamma^*(x,y)\big(\phi_{\theta,\omega}(x) + \psi_\theta(y) - \|x-y\|^2/2\big) dx dy + \|\gamma_{\theta,\omega}\|_1 - \|\gamma^*\|_1 =$$

$$\varepsilon^{-1} \int_{\mathbb{R}^d \times \mathbb{R}^d} \gamma^*(x,y)\big(\phi^*(x) + \psi^*(y) - \phi_{\theta,\omega}(x) - \psi_\theta(y)\big) dx dy + \|\gamma_{\theta,\omega}\|_1 - \|\gamma^*\|_1 =$$

$$\varepsilon^{-1} \int_{\mathbb{R}^d} \frac{\gamma_x^*(x)}{p(x)}\big(\phi^*(x) - \phi_{\theta,\omega}(x)\big) p(x) dx +$$

$$\varepsilon^{-1} \int_{\mathbb{R}^d} \frac{\gamma_y^*(y)}{q(y)}\big(\psi^*(y) - \psi_\theta(y)\big) q(y) dy + \|\gamma_{\theta,\omega}\|_1 - \|\gamma^*\|_1 \stackrel{(26)-(27)}{\leq}$$

$$\varepsilon^{-1} \int_{\mathbb{R}^d} (\overline{f}_1(-\phi_{\theta,\omega}(x)) - \overline{f}_1(-\phi^*(x))) p(x) dx +$$

$$\varepsilon^{-1} \int_{\mathbb{R}^d} (\overline{f}_2(-\psi_\theta(y)) - \overline{f}_2(-\psi^*(y))) q(y) dy + \|\gamma_{\theta,\omega}\|_1 - \|\gamma^*\|_1 =$$

$$\varepsilon^{-1} \cdot \left( \int_{\mathbb{R}^d} \overline{f}_1(-\phi_{\theta,\omega}(x)) p(x) dx + \int_{\mathbb{R}^d} \overline{f}_2(-\psi_\theta(y)) q(y) dy + \varepsilon\|\gamma_{\theta,\omega}\|_1 - \right.$$

$$\left. \underbrace{(\int_{\mathbb{R}^d} \overline{f}_1(-\phi^*(x)) p(x) dx + \int_{\mathbb{R}^d} \overline{f}_2(-\psi^*(y)) q(y) dy + \varepsilon\|\gamma^*\|_1)}_{\mathcal{L}^* \stackrel{\text{def}}{=}} \right) =$$

$$\varepsilon^{-1}(\mathcal{L}(\theta,\omega) - \mathcal{L}^*).$$

$\square$

## A.2 Proof of Proposition 4.2

We begin with proving the auxiliary theoretical results (Propositions A.1, A.2) which are needed to prove the main proposition of this section.

**Proposition A.1** (Rademacher bound on the estimation error)**.** *It holds that*

$$\mathbb{E}\big[\mathcal{L}(\widehat{\theta}) - \mathcal{L}(\overline{\theta})\big] \leq 4\mathcal{R}_N(\mathcal{F}_1, p) + 4\mathcal{R}_M(\mathcal{F}_2, q),$$

*where* $\mathcal{F}_1 = \{\overline{f}_1(-\phi_{\theta,\omega})|(\theta,\omega) \in \Theta \times \Omega\}$, $\mathcal{F}_2 = \{\overline{f}_2(-\psi_\theta)|\theta \in \Theta\}$ *for* $\phi_{\theta,\omega}(x) = \varepsilon \log \frac{u_\omega(x)}{c_\theta(x)} + \frac{\|x\|^2}{2}$, $\psi_\theta(y) = \varepsilon \log v_\theta(y) + \frac{\|y\|^2}{2}$, *and* $\mathcal{R}_N(\mathcal{F}_1, p)$, $\mathcal{R}_M(\mathcal{F}_2, q)$ *denote the Rademacher complexity* [63, §26] *of the functional classes* $\mathcal{U}$, $\mathcal{V}$ *w.r.t. to the sample sizes* $N$, $M$ *of distributions* $p$, $q$.

*Proof of Proposition A.1.* The derivation of this fact is absolutely analogous to [41, Proposition B.1], [50, Theorem 4] or [64, Theorem 3.4]. ☐

**Proposition A.2** (Bound on the Rademacher complexity of the considered classes)**.** *Let* $0 < a \leq A$, *let* $0 < u \leq U$, *let* $0 < w \leq W$ *and* $V > 0$. *Consider the class of functions*

$$\mathcal{F}_1 = \big\{x \mapsto \overline{f}_1(-\varepsilon \log u_\omega(x) + \varepsilon \log c_\theta(x) - \frac{\|x\|^2}{2})\big\},$$

$$\mathcal{F}_2 = \big\{y \mapsto \overline{f}_2(-\varepsilon \log v_\theta(y) - \frac{\|y\|^2}{2})\big\} \text{ where}$$

$$u_\omega(x), \ v_\theta(y), \ c_\theta(x) \text{ belong to the class}$$

$$\mathcal{V} = \Big\{x \mapsto \sum_{k=1}^{K} \alpha_k \exp\big(x^T U_k x + v_k^T x + w_k\big) \text{ with} \tag{29}$$

$$uI \preceq U_k = U_k^T \preceq UI; \|v_k\| \leq V; w \leq w_k \leq W; a \leq \alpha_k \leq A\Big\}.$$

*Following* [41, Proposition B.2], *we call the functions of the class* $\mathcal{V}$ *as constrained log-sum-exp quadratic functions. We assume that* $\overline{f}_1$, $\overline{f}_2$ *are Lipshitz functions and measures* $p$, $q$ *are compactly supported with the supports lying in a zero-centered ball of a radius* $R > 0$. *Then*

$$\mathcal{R}_N(\mathcal{F}_1, p) \leq \frac{C_0}{\sqrt{N}}, \ \mathcal{R}_M(\mathcal{F}_2, q) \leq \frac{C_1}{\sqrt{M}}$$

*where the constants* $C_0$, $C_1$ ***do not depend*** *on sizes* $N$, $M$ *of the empirical samples from* $p$, $q$.

*Proof of Proposition A.2.* Thanks to [41, Proposition B.2], the Rademacher complexities of constrained log-sum-exp quadratic functions $x \mapsto \log u_\omega(x)$, $x \mapsto \log c_\theta(x)$ and $y \mapsto \log v_\theta(y)$ are known to be bounded by $O(\frac{1}{\sqrt{N}})$ or $O(\frac{1}{\sqrt{M}})$ respectively. According to the definition of Rademacher complexity, for single quadratic functions $x \mapsto \frac{x^T x}{2}$ ($y \mapsto \frac{y^T y}{2}$) it is just equal to zero. Then, using the well-known scaling and additivity properties of the Rademacher complexity [63], we get that $x \mapsto -\varepsilon \log u_\omega(x) + \varepsilon \log c_\theta(x) - \frac{\|x\|^2}{2}$ and $y \mapsto -\varepsilon \log v_\theta(y) - \frac{\|y\|^2}{2}$ are bounded by $O(\frac{1}{\sqrt{N}})$ and $O(\frac{1}{\sqrt{M}})$ respectively. The remaining step is to recall that $\overline{f}_1(x)$ and $\overline{f}_2(y)$ are Lipschitz. Therefore, according to Talagrand's contraction principle [49], the Rademaher complexities of $\mathcal{F}_1$ and $\mathcal{F}_2$ are also bounded by $O(\frac{1}{\sqrt{N}})$ and $O(\frac{1}{\sqrt{M}})$, respectively. ☐

*Proof of Proposition 4.2.* The proof directly follows from Propositions A.1 and A.2. ☐

## A.3 Proof of Theorem 4.3

To begin with, we provide a quick reminder about the Fenchel-Rockafellar theorem which is needed to derive the dual form of problem (3).

**Theorem A.3** (Fenchel-Rockafellar [58])**.** *Let* $(E, E')$ *and* $(F, F')$ *be two couples of topologically paired spaces. Let* $A : E \mapsto F$ *be a continuous linear operator and* $\overline{A} : F' \mapsto E'$ *be its adjoint. Let*

*f* and *g* be lower semi-continuous and proper convex functions defined on $E$ and $F$ respectively. If there exists $x \in \mathrm{dom} f$ s.t. $g$ is continuous at $Ax$, then

$$\sup_{x \in E} -f(-x) - g(Ax) = \min_{\overline{y} \in F'} \overline{f}(\overline{A}\,\overline{y}) + \overline{g}(\overline{y})$$

and the min is attained. Moreover, if there exists a maximizer $x \in E$ then there exists $\overline{y} \in F'$ satisfying $Ax \in \partial \overline{g}(\overline{y})$ and $\overline{A}\overline{y} \in \partial f(-x)$.

We note that in the below theorem $\mathcal{C}_2(\mathbb{R}^d)$ **does not** denote the space of twice differentiable continuous functions. The exact definition of this space and $\mathcal{C}_{2,b}(\mathbb{R}^d)$ is given in **notations** part.

**Theorem A.4** (Dual form of problem (3)). *The primal UEOT problem* (3) *has the dual counterpart* (4) *where the potentials* $(\phi, \psi)$ *belong to the space* $\mathcal{C}_{2,b}(\mathbb{R}^d) \times \mathcal{C}_{2,b}(\mathbb{R}^d)$. *The minimum of* (3) *is attained for a unique* $\gamma^* \in \mathcal{M}_{2,+}(\mathbb{R}^d \times \mathbb{R}^d)$.

*Proof.* We recall that in the primal form, the minimization is performed over functions $\gamma$ belonging to $\mathcal{M}_{2,+}(\mathbb{R}^d \times \mathbb{R}^d)$. In this proof, we suppose that this space is endowed with a coarsest topology $\sigma(\mathcal{M}_{2,+}(\mathbb{R}^d \times \mathbb{R}^d))$ which makes continuous the linear functionals $\gamma \mapsto \int \zeta d\gamma$, $\forall \zeta \in \mathcal{C}_2(\mathbb{R}^d \times \mathbb{R}^d)$. Then the topological space $\left(\mathcal{M}_{2,+}(\mathbb{R}^d \times \mathbb{R}^d), \sigma(\mathcal{M}_{2,+}(\mathbb{R}^d \times \mathbb{R}^d))\right)$ has a topological dual $\left(\mathcal{M}_{2,+}(\mathbb{R}^d \times \mathbb{R}^d), \sigma(\mathcal{M}_{2,+}(\mathbb{R}^d \times \mathbb{R}^d))\right)'$ which, actually, is (linear) isomorphic to the space $\mathcal{C}_2(\mathbb{R}^d \times \mathbb{R}^d)$, see [26, Lemma 9.9]. This fact opens an opportunity to apply the well-celebrated Fenchel-Rockafellar theorem. For this purpose, we will consider the following spaces: $E \overset{\mathrm{def}}{=} \mathcal{C}_2(\mathbb{R}^d) \times \mathcal{C}_2(\mathbb{R}^d)$, $F \overset{\mathrm{def}}{=} \mathcal{C}_2(\mathbb{R}^d \times \mathbb{R}^d)$ and their duals $E' \overset{\mathrm{def}}{=} \mathcal{M}_{2,+}(\mathbb{R}^d) \times \mathcal{M}_{2,+}(\mathbb{R}^d)$ and $F' \overset{\mathrm{def}}{=} \mathcal{M}_{2,+}(\mathbb{R}^d \times \mathbb{R}^d)$.

*Step 1.* Recall that the convex conjugate of any function $g : \mathcal{M}_{2,+}(\mathbb{R}^d \times \mathbb{R}^d) \to \mathbb{R} \cup \{+\infty\}$ is defined for each $\zeta \in (\mathcal{M}_{2,+}(\mathbb{R}^d \times \mathbb{R}^d))' \cong \mathcal{C}_2(\mathbb{R}^d \times \mathbb{R}^d)$ as $\overline{g}(\zeta) = \sup_{\gamma \in \mathcal{M}_{2,+}(\mathbb{R}^d \times \mathbb{R}^d)} \{\langle \gamma, \zeta \rangle - g(\gamma)\}$. For the convenience of further derivations, we introduce additional functionals corresponding to the summands in the primal UEOT problem (3):

$$P(\gamma) \overset{\mathrm{def}}{=} \int_{\mathbb{R}^d} \int_{\mathbb{R}^d} \frac{\|x - y\|^2}{2} \gamma(x,y) dx dy - \varepsilon H(\gamma);$$

$$F_1(\gamma_x) \overset{\mathrm{def}}{=} D_{f_1}(\gamma_x \| p); \quad F_2(\gamma_y) \overset{\mathrm{def}}{=} D_{f_2}(\gamma_y \| q). \tag{30}$$

For our purposes, we need to calculate the convex conjugates of these functionals. Fortunately, convex conjugates of $f$-divergences $F_1(\gamma_x)$ and $F_2(\gamma_y)$ are well-known, see [1, Proposition 23], and equal to

$$\overline{F_1}(\phi) \overset{\mathrm{def}}{=} \int_{\mathbb{R}^d} \overline{f_1}(\phi(x)) p(x) dx, \quad \overline{F_2}(\psi) \overset{\mathrm{def}}{=} \int_{\mathbb{R}^d} \overline{f_2}(\psi(y)) q(y) dy.$$

To proceed, we calculate the convex conjugate of $P(\gamma)$:

$$\overline{P}(\zeta) = \overline{\int_{\mathbb{R}^d} \int_{\mathbb{R}^d} \frac{\|x - y\|^2}{2} \gamma(x,y) dx dy - \varepsilon H(\gamma)} =$$

$$= \sup_{\gamma \in \mathcal{M}_{2,+}(\mathbb{R}^d \times \mathbb{R}^d)} \left\{ \int_{\mathbb{R}^d} \int_{\mathbb{R}^d} \zeta(x,y) \gamma(x,y) dx dy - \Big( \int_{\mathbb{R}^d} \int_{\mathbb{R}^d} \frac{\|x - y\|^2}{2} \gamma(x,y) dx dy - \right.$$

$$\left. \varepsilon H(\gamma) \Big) \right\} =$$

$$\sup_{\gamma \in \mathcal{M}_{2,+}(\mathbb{R}^d \times \mathbb{R}^d)} \left\{ \int_{\mathbb{R}^d} \int_{\mathbb{R}^d} \big( \zeta(x,y) - \frac{\|x - y\|^2}{2} \big) \gamma(x,y) dx dy + \right.$$

$$\left. \varepsilon \underbrace{\Big( \int_{\mathbb{R}^d} \int_{\mathbb{R}^d} -\gamma(x,y) \log \gamma(x,y) dx dy + \|\gamma\|_1 \Big)}_{= H(\gamma)} \right\} =$$

$$\sup_{\gamma \in \mathcal{M}_{2,+}(\mathbb{R}^d \times \mathbb{R}^d)} \varepsilon \cdot \left\{ \int_{\mathbb{R}^d} \int_{\mathbb{R}^d} \gamma(x,y) \Big( \frac{\zeta(x,y) - \frac{\|x-y\|^2}{2}}{\varepsilon} - \log \gamma(x,y) \Big) dx dy + \|\gamma\|_1 \right\} =$$

$$\sup_{\gamma \in \mathcal{M}_{2,+}(\mathbb{R}^d \times \mathbb{R}^d)} (-\varepsilon) \cdot \left\{ \int_{\mathbb{R}^d} \int_{\mathbb{R}^d} \gamma(x,y) \left( \log \gamma(x,y) - \frac{\zeta(x,y) - \frac{\|x-y\|^2}{2}}{\varepsilon} \right) dxdy - \|\gamma\|_1 \right\} =$$

$$\sup_{\gamma \in \mathcal{M}_{2,+}(\mathbb{R}^d \times \mathbb{R}^d)} (-\varepsilon) \cdot \left\{ \int_{\mathbb{R}^d} \int_{\mathbb{R}^d} \gamma(x,y) \left( \log \gamma(x,y) - \frac{\zeta(x,y) - \frac{\|x-y\|^2}{2}}{\varepsilon} \right) dxdy - \|\gamma\|_1 + \right.$$

$$\left. \underbrace{\int_{\mathbb{R}^d} \int_{\mathbb{R}^d} \exp\left\{ \frac{\zeta(x,y) - \frac{\|x-y\|^2}{2}}{\varepsilon} \right\} dxdy - \int_{\mathbb{R}^d} \int_{\mathbb{R}^d} \exp\left\{ \frac{\zeta(x,y) - \frac{\|x-y\|^2}{2}}{\varepsilon} \right\} dxdy}_{=0} \right\} = \quad (31)$$

$$\sup_{\gamma \in \mathcal{M}_{2,+}(\mathbb{R}^d \times \mathbb{R}^d)} (-\varepsilon) \cdot \left\{ \mathrm{D_{KL}} \left( \gamma \| \exp\left\{ \frac{\zeta(x,y) - \frac{\|x-y\|^2}{2}}{\varepsilon} \right\} \right) - \right.$$

$$\left. \int_{\mathbb{R}^d} \int_{\mathbb{R}^d} \exp\left\{ \frac{\zeta(x,y) - \frac{\|x-y\|^2}{2}}{\varepsilon} \right\} dxdy \right\}. \quad (32)$$

Here in the transition from (31) to (32), we keep in mind our prior calculations of $\mathrm{D_{KL}}$ in (28). Recall that $\mathrm{D_{KL}}$ is non-negative and attains zero at the unique point

$$\gamma(x,y) = \exp\left\{ \frac{\zeta(x,y) - \frac{\|x-y\|^2}{2}}{\varepsilon} \right\}. \quad (33)$$

Thus, we get

$$\overline{P}(\zeta) = \varepsilon \int_{\mathbb{R}^d} \int_{\mathbb{R}^d} \exp\left\{ \frac{\zeta(x,y) - \frac{\|x-y\|^2}{2}}{\varepsilon} \right\} dxdy. \quad (34)$$

*Step 2.* Now we are ready to apply the Fenchel-Rockafellar theorem in our case. To begin with, we show that this theorem is applicable to problem (3), i.e., that the functions under consideration satisfy the necessary conditions. Indeed, it is known that the convex conjugate of any functional (e.g., $\overline{F_1}(\cdot)$, $\overline{F_2}(\cdot)$, $\overline{P}(\cdot)$) is **lower semi-continuous** and **convex**. Besides, the listed functionals are **proper convex**[2]. Indeed, the properness of $\overline{F_1}(\cdot)$ and $\overline{F_2}(\cdot)$ follows from the fact that $f$-divergences are known to be lower-semicontinuous and proper themselves, while properness of $\overline{P}(\cdot)$ is evident from (32).

Now we consider the linear operator $A : \mathcal{C}_2(\mathbb{R}^d) \times \mathcal{C}_2(\mathbb{R}^d) \mapsto \mathcal{C}_2(\mathbb{R}^d \times \mathbb{R}^d)$ which is defined as $A(\phi,\psi) : (x,y) \mapsto \phi(x) + \psi(y)$. It is continuous, and its adjoint is defined on $\mathcal{M}_{2,+}(\mathbb{R}^d \times \mathbb{R}^d)$ as $\overline{A}(\gamma) = (\gamma_x, \gamma_y)$. Indeed, $\langle \overline{A}(\gamma), (u,v) \rangle = \langle \gamma, A(u,v) \rangle = \int_{\mathbb{R}^d} \int_{\mathbb{R}^d} \gamma(x,y)(u(x) + v(y)) dxdy = \int_{\mathbb{R}^d} \gamma_x(x)u(x)dx + \int_{\mathbb{R}^d} \gamma_y(y)v(y)dy$. Thus, the strong duality and the existence of minimizer for (3) follows from the Fenchel-Rockafellar theorem which states that problems

$$\sup_{(\phi,\psi) \in \mathcal{C}_2(\mathbb{R}^d) \times \mathcal{C}_2(\mathbb{R}^d)} \left\{ -\overline{P}(A(\phi,\psi)) - \overline{F_1}(-\phi) - \overline{F_2}(-\psi) \right\} \quad (35)$$

and

$$\min_{\gamma \in \mathcal{M}_{2,+}(\mathbb{R}^d \times \mathbb{R}^d)} \left\{ P(\gamma) + F_1(\gamma_x) + F_2(\gamma_y) \right\} \quad (36)$$

are equal. The uniqueness of the minimizer for (3) comes from the strict convexity of $P(\cdot)$ (which holds thanks to the entropy term). Note that the conjugate of the sum of $F_1$ and $F_2$ is equal to the sum of their conjugates since they are defined for separate non-intersecting groups of parameters.

Next we prove that the supremum can be restricted to $\mathcal{C}_{2,b}(\mathbb{R}^d \times \mathbb{R}^d)$. Here we use "$\wedge$" to denote the operation of taking minimum between the function $f : \mathbb{R}^d \to \mathbb{R}$ and real value $k$: $(f \wedge k)(x) \stackrel{\text{def}}{=} \min(f(x), k)$. Then analogously to [26, Theorem 9.6], we get:

---

[2]*Proper convex* function is a real-valued convex function which has a non-empty domain, never attains the value $(-\infty)$ and is not identically equal to $(+\infty)$. This property ensures that the minimization problem for this function has non-trivial solutions.

$$\sup_{(\phi,\psi)\in\mathcal{C}_2(\mathbb{R}^d)\times\mathcal{C}_2(\mathbb{R}^d)} \{-\overline{P}(A(\phi,\psi))-\overline{F_1}(-\phi)-\overline{F_2}(-\psi)\}=$$

$$\sup_{(\phi,\psi)\in\mathcal{C}_2(\mathbb{R}^d)\times\mathcal{C}_2(\mathbb{R}^d)}\lim_{k_1,k_2\to\infty}\{-\overline{P}(A(\phi\wedge k_1,\psi\wedge k_2))-\overline{F_1}(-(\phi\wedge k_1))-\overline{F_2}(-(\psi\wedge k_2))\}\leq$$

$$\sup_{(\phi,\psi)\in\mathcal{C}_{2,b}(\mathbb{R}^d)\times\mathcal{C}_{2,b}(\mathbb{R}^d)}\{-\overline{P}(A(\phi,\psi))-\overline{F_1}(-\phi)-\overline{F_2}(-\psi)\}. \quad (37)$$

Since the another inequality is obvious, the two quantities are equal which completes the proof.

$\square$

*Proof of Theorem 4.3.* Our aim is to prove that for all $\delta > 0$ there exist unnormalized Gaussian mixtures $u_\omega$ and $v_\theta$ s.t. $\mathcal{L}(\theta,\omega)-\mathcal{L}^* < \delta\varepsilon$. We define

$$\mathcal{J}(\phi,\psi)\stackrel{\text{def}}{=}$$
$$\varepsilon\int_{\mathbb{R}^d}\int_{\mathbb{R}^d}\exp\{\frac{1}{\varepsilon}(\phi(x)+\psi(y)-\frac{\|x-y\|^2}{2})\}dxdy+\int_{\mathbb{R}^d}\overline{f}_1(-\phi(x))p(x)dx+\int_{\mathbb{R}^d}\overline{f}_2(-\psi(y))q(y)dy.$$

Then from (4) and Theorem (A.4), it follows that $\mathcal{L}^* = \inf_{(\phi,\psi)\in\mathcal{C}_{2,b}(\mathbb{R}^d)\times\mathcal{C}_{2,b}(\mathbb{R}^d)}\mathcal{J}(\phi,\psi)$. Finally, using the definition of the infimum, we get that for all $\delta' > 0$ there exist some functions $(\widehat{\phi},\ \widehat{\psi})\in\mathcal{C}_{2,b}(\mathbb{R}^d)\times\mathcal{C}_{2,b}(\mathbb{R}^d)$ such that $\mathcal{J}(\widehat{\phi},\widehat{\psi})<\mathcal{L}^*+\delta'$. For further derivations, we set $\delta'\stackrel{\text{def}}{=}\frac{\delta\varepsilon}{2}$ and pick the corresponding $(\widehat{\phi},\ \widehat{\psi})$.

*Step 1.* We start with the derivation of some inequalities useful for future steps. Since $(\widehat{\phi},\widehat{\psi})\in\mathcal{C}_{2,b}(\mathbb{R}^d)\times\mathcal{C}_{2,b}(\mathbb{R}^d)$, they have upper bounds $\widehat{a}$ and $\widehat{b}$ such that for all $x,y\in\mathbb{R}^d$: $\widehat{\phi}(x)\leq\widehat{a}$ and $\widehat{\psi}(y)\leq\widehat{b}$ respectively. We recall that by the assumption of the theorem, measures $p$ and $q$ are compactly supported. Thus, there exist balls centered at $x=0$ and $y=0$ and having some radius $R>0$ which contain the supports of $p$ and $q$ respectively. Then we define

$$\widetilde{\phi}(x)\stackrel{\text{def}}{=}\widehat{\phi}(x)-\max\{0,\max\{\|x\|^2-R^2,\|x\|^4-R^4\}\}\leq\widehat{\phi}(x)\leq\widehat{a};$$
$$\widetilde{\psi}(y)\stackrel{\text{def}}{=}\widehat{\psi}(y)-\max\{0,\|y\|^2-R^2\}\leq\widehat{\psi}(y)\leq\widehat{b}.$$

We get that

$$\widetilde{\phi}(x)\leq\widehat{\phi}(x),\widetilde{\psi}(y)\leq\widehat{\psi}(y)\Longrightarrow$$
$$\varepsilon\int_{\mathbb{R}^d}\int_{\mathbb{R}^d}\exp\{\frac{1}{\varepsilon}(\widetilde{\phi}(x)+\widetilde{\psi}(y)-\frac{\|x-y\|^2}{2})\}dxdy\leq$$
$$\varepsilon\int_{\mathbb{R}^d}\int_{\mathbb{R}^d}\exp\{\frac{1}{\varepsilon}(\widehat{\phi}(x)+\widehat{\psi}(y)-\frac{\|x-y\|^2}{2})\}dxdy \quad (38)$$

Importantly, for all $x$ and $y$ within the supports of $p$ and $q$ it holds that $\widetilde{\phi}(x)=\widehat{\phi}(x)$ and $\widetilde{\psi}(y)=\widehat{\psi}(y)$, respectively. Then

$$\int_{\mathbb{R}^d}\overline{f}_1(-\widehat{\phi}(x))p(x)dx=\int_{\mathbb{R}^d}\overline{f}_1(-\widetilde{\phi}(x))p(x)dx,$$
$$\int_{\mathbb{R}^d}\overline{f}_2(-\widehat{\psi}(y))q(y)dy=\int_{\mathbb{R}^d}\overline{f}_2(-\widetilde{\psi}(y))q(y)dy. \quad (39)$$

Combining (38) and (39), we get that $\mathcal{J}(\widetilde{\phi},\widetilde{\psi})\leq\mathcal{J}(\widehat{\phi},\widehat{\psi})<\mathcal{L}^*+\delta$.

Before moving on, we note that functions $\exp\{\widetilde{\phi}(x)/\varepsilon\}$ and $\exp\{\widetilde{\psi}(y)/\varepsilon\}$ are continuous and non-negative. Therefore, since measures $p$ and $q$ are compactly supported, there exist some constants $e_{\min},\ h_{\min}>0$ such that $\exp\{\widetilde{\phi}(x)/\varepsilon\}>e_{\min}$ and $\exp\{\widetilde{\psi}(y)/\varepsilon\}>h_{\min}$ for all $x$ and $y$ from the supports of measures $p$ and $q$ respectively. We keep these constants for future steps.

*Step 2.* This step of our proof is similar to [41, Theorem 3.4]. We get that

$$\exp\left(\widetilde{\phi}(x)/\varepsilon\right)\leq\exp\left(\frac{\widehat{a}-\max\{0,\|x\|^2-R^2\}}{\varepsilon}\right)\leq\exp\left(\frac{\widehat{a}+R^2}{\varepsilon}\right)\cdot\exp(-\|x\|^2/\varepsilon), \quad (40)$$

$$\exp\left(\widetilde{\psi}(y)/\varepsilon\right) \le \exp\left(\frac{\widehat{b} - \max\{0, \|y\|^2 - R^2\}}{\varepsilon}\right) \le \exp\left(\frac{\widehat{b} + R^2}{\varepsilon}\right) \cdot \exp(-\|y\|^2/\varepsilon).$$

From this we can deduce that $y \mapsto \exp(\widetilde{\psi}(y)/\varepsilon)$ is a normalizable density since it is bounded by the unnormalized Gaussian density. Moreover, we see that it vanishes at the infinity. Thus, using the result [52, Theorem 5a], we get that for all $\delta'' > 0$ there exists an unnormalized Gaussian mixture $v_{\widetilde{\theta}} = v_{\widetilde{\theta}}(y)$ such that

$$\|v_{\widetilde{\theta}} - \exp(\widetilde{\psi}/\varepsilon)\|_\infty = \sup_{y \in \mathbb{R}^D} |v_{\widetilde{\theta}}(y) - \exp(\widetilde{\psi}(y)/\varepsilon)| < \delta''. \tag{41}$$

Following the mentioned theorem, we can set all the covariances in $v_{\widetilde{\theta}}$ to be scalar, i.e., define $v_{\widetilde{\theta}}(y) = \sum_{k=1}^K \widetilde{\alpha}_k \mathcal{N}(y|\widetilde{r}_k, \varepsilon\widetilde{\lambda}_k I_d)$ for some $K$ and $\widetilde{\alpha}_k \in \mathbb{R}_+, \widetilde{r}_k \in \mathbb{R}^d, \widetilde{\lambda}_k \in \mathbb{R}_+$ ($k \in \{1, \ldots, K\}$). For our future needs, we set

$$\delta'' = \frac{\delta\varepsilon}{2} \cdot \left[L_1 \cdot \frac{\varepsilon}{e_{\min}} + L_2 \cdot \frac{\varepsilon}{h_{\min}} + \varepsilon(2\pi\varepsilon)^{\frac{d}{2}}\left(1 + (\pi\varepsilon)^{\frac{d}{2}}\exp\left\{\frac{\widehat{a} + R^2}{\varepsilon}\right\}\right)\right]^{-1}.$$

For simplicity, we consider the other mixture $v_\theta(y) \stackrel{\text{def}}{=} v_{\widetilde{\theta}}(y)\exp(-\frac{\|y\|^2}{2\varepsilon})$ which is again unnormalized and has scalar covariances, see the proof of [41, Theorem 3.4] for explanation. We denote the weights, means, and covariances of this mixture by $\alpha_k \in \mathbb{R}_+, r_k \in \mathbb{R}^d$ and $\lambda_k \in \mathbb{R}_+$, respectively.

We derive that

$$\varepsilon\int_{\mathbb{R}^d}\int_{\mathbb{R}^d}\exp\left\{\frac{1}{\varepsilon}(\widetilde{\phi}(x) + \widetilde{\psi}(y) - \frac{\|x-y\|^2}{2})\right\}dxdy =$$

$$\varepsilon\int_{\mathbb{R}^d}\int_{\mathbb{R}^d}\exp\{\frac{\widetilde{\phi}(x)}{\varepsilon}\}\exp\{-\frac{\|x-y\|^2}{2\varepsilon}\}\exp\{\frac{\widetilde{\psi}(y)}{\varepsilon}\}dxdy >$$

$$\varepsilon\int_{\mathbb{R}^d}\int_{\mathbb{R}^d}\exp\{\frac{\widetilde{\phi}(x)}{\varepsilon}\}\exp\{-\frac{\|x-y\|^2}{2\varepsilon}\}(v_{\widetilde{\theta}}(y) - \delta'')dxdy =$$

$$\varepsilon\int_{\mathbb{R}^d}\int_{\mathbb{R}^d}\exp\{\frac{\widetilde{\phi}(x)}{\varepsilon}\}\exp\{-\frac{\|x-y\|^2}{2\varepsilon}\}v_{\widetilde{\theta}}(y)dxdy -$$

$$\delta''\varepsilon\int_{\mathbb{R}^d}\int_{\mathbb{R}^d}\exp\{\frac{\widetilde{\phi}(x)}{\varepsilon}\}\exp\{-\frac{\|x-y\|^2}{2\varepsilon}\}dxdy =$$

$$\varepsilon\int_{\mathbb{R}^d}\int_{\mathbb{R}^d}\exp\{\frac{\widetilde{\phi}(x)}{\varepsilon}\}\exp\{-\frac{\|x\|^2}{2\varepsilon}\}\exp\{\frac{\langle x,y\rangle}{\varepsilon}\}\underbrace{\exp\{-\frac{\|y\|^2}{2\varepsilon}\}v_{\widetilde{\theta}}(y)}_{=v_\theta(y)}dxdy -$$

$$\delta''\varepsilon\int_{\mathbb{R}^d}\int_{\mathbb{R}^d}\exp\{\frac{\widetilde{\phi}(x)}{\varepsilon}\}\exp\{-\frac{\|x-y\|^2}{2\varepsilon}\}dxdy =$$

$$\varepsilon\int_{\mathbb{R}^d}\exp\{\frac{\widetilde{\phi}(x)}{\varepsilon}\}\exp\{-\frac{\|x^2\|}{2\varepsilon}\}\left(\underbrace{\int_{\mathbb{R}^d}\exp\{\frac{\langle x,y\rangle}{\varepsilon}\}v_\theta(y)dy}_{=c_\theta(x)}\right)dx -$$

$$\delta''\varepsilon\int_{\mathbb{R}^d}\exp\{\frac{\widetilde{\phi}(x)}{\varepsilon}\}\left(\underbrace{\int_{\mathbb{R}^d}\exp\{-\frac{\|x-y\|^2}{2\varepsilon}\}dy}_{=(2\pi\varepsilon)^{d/2}}\right)dx =$$

$$\varepsilon\int_{\mathbb{R}^d}\exp\{\frac{\widetilde{\phi}(x)}{\varepsilon}\}\exp\{-\frac{\|x^2\|}{2\varepsilon}\}c_\theta(x)dx - \delta''\varepsilon(2\pi\varepsilon)^{d/2}\int_{\mathbb{R}^d}\exp\{\frac{\widetilde{\phi}(x)}{\varepsilon}\}dx \stackrel{(40)}{>}$$

$$\varepsilon\int_{\mathbb{R}^d}\exp\{\frac{\widetilde{\phi}(x)}{\varepsilon}\}\exp\{-\frac{\|x^2\|}{2\varepsilon}\}c_\theta(x)dx -$$

$$\delta'' \varepsilon (2\pi\varepsilon)^{d/2} \exp\left\{\frac{\widehat{a} + R^2}{\varepsilon}\right\} \int_{\mathbb{R}^d} \underbrace{\exp\{-\|x\|^2/\varepsilon\} dx}_{=(\pi\varepsilon)^{d/2}} =$$

$$\varepsilon \int_{\mathbb{R}^d} \exp\{\frac{\widetilde{\phi}(x)}{\varepsilon}\} \exp\{-\frac{\|x^2\|}{2\varepsilon}\} c_\theta(x) dx - \delta'' 2^{d/2} \pi^d \varepsilon^{(d+1)} \exp\left\{\frac{\widehat{a} + R^2}{\varepsilon}\right\}. \tag{42}$$

_Step 3._ At this point, we will show that for every $\delta'' > 0$, there exists an unnormalized Gaussian mixture $u_{\widetilde{\omega}}$ which is $\delta''$-close to $\exp\{\widetilde{\phi}(x)/\varepsilon\} c_\theta(x)$. Using the closed-form expression for $c_{\theta(x)}$ from [41, Proposition 3.2], we get that

$$c_\theta(x) = \sum_{k=1}^{K} \alpha_k \exp\{-\frac{\|r_k\|^2}{2\varepsilon\lambda_k}\} \exp\{\frac{\|r_k + x\lambda_k\|^2}{2\varepsilon\lambda_k}\}.$$

Then

$$\exp\left(\widetilde{\phi}(x)/\varepsilon\right) c_\theta(x) \leq$$

$$\exp\left(\frac{\widehat{a} - \max\{0, \|x\|^4 - R^4\}}{\varepsilon}\right) c_\theta(x) \leq \exp\left(\frac{\widehat{a} + R^4}{\varepsilon}\right) \cdot \exp(-\|x\|^4/\varepsilon) c_\theta(x) =$$

$$\sum_{k=1}^{K} \alpha_k \exp\left(\frac{\widehat{a} + R^4}{\varepsilon}\right) \exp\{-\frac{\|r_k\|^2}{2\varepsilon\lambda_k}\} \cdot \exp(-\frac{\|x\|^4}{\varepsilon}) \exp\{\frac{\|r_k + x\lambda_k\|^2}{2\varepsilon\lambda_k}\} =$$

$$\sum_{k=1}^{K} \alpha_k \exp\left(\frac{\widehat{a} + R^4}{\varepsilon}\right) \exp\{-\frac{\|r_k\|^2}{2\varepsilon\lambda_k}\} \cdot \exp\{\frac{\|r_k + x\lambda_k\|^2 - 2\lambda_k\|x\|^4}{2\varepsilon\lambda_k}\}$$

From this, we see that $\exp\left(\widetilde{\phi}(x)/\varepsilon\right) c_\theta(x)$ tends to zero while $x$ approaches infinity. It means that $x \mapsto \exp\left(\widetilde{\phi}(x)/\varepsilon\right) c_\theta(x)$ corresponds to the normalizable density. Using [52, Theorem 5a], we get that for all $\delta'' > 0$ there exists an unnormalized Gaussian mixture $u_{\widetilde{\omega}}$ such that

$$\|u_{\widetilde{\omega}} - \exp(\widetilde{\phi}/\varepsilon) c_\theta\|_\infty = \sup_{x \in \mathbb{R}^D} |u_{\widetilde{\omega}}(x) - \exp(\widetilde{\phi}(x)/\varepsilon) c_\theta(x))| < \delta''. \tag{43}$$

Analogously with $v_{\widetilde{\theta}}$, we can set all the covariances in $u_{\widetilde{\omega}}$ to be scalar, i.e., define $u_{\widetilde{\omega}} = \sum_{l=1}^{L} \widetilde{\beta}_l \mathcal{N}(x | \widetilde{\mu}_l, \varepsilon\widetilde{\zeta}_l I_d)$ for some $L$, $\widetilde{\mu}_l \in \mathbb{R}^d$, $\widetilde{\zeta}_l \in \mathbb{R}_+$ ($l \in \{1, ..., L\}$). Moreover, we consider $u_\omega(x) = u_{\widetilde{\omega}}(x) \exp\{-\frac{\|x\|^2}{2\varepsilon}\}$ which is again an unnormalized density with scalar covariances. We denote the weights, means, covariances of this mixture by $\beta_l \in \mathbb{R}_+$, $\mu_l \in \mathbb{R}^d$, $\zeta_l \in \mathbb{R}_+$ respectively.

Next we recall the equation (42) and get that

$$(42) > \varepsilon \int_{\mathbb{R}^d} \exp\{-\frac{\|x\|^2}{2\varepsilon}\}(u_{\widetilde{\omega}}(x) - \delta'') dx - \delta'' 2^{d/2} \pi^d \varepsilon^{(d+1)} \exp\left\{\frac{\widehat{a} + R^2}{\varepsilon}\right\} =$$

$$\varepsilon \int_{\mathbb{R}^d} \underbrace{\exp\{-\frac{\|x\|^2}{2\varepsilon}\} u_{\widetilde{\omega}}(x)}_{=u_\omega(x)} dx - \varepsilon\delta'' \underbrace{\int_{\mathbb{R}^d} \exp\{-\frac{\|x\|^2}{2\varepsilon}\} dx}_{=(2\pi\varepsilon)^{\frac{d}{2}}} - \delta'' 2^{d/2} \pi^d \varepsilon^{(d+1)} \exp\left\{\frac{\widehat{a} + R^2}{\varepsilon}\right\} =$$

$$\varepsilon\|u_\omega\|_1 - \varepsilon\delta''(2\pi\varepsilon)^{\frac{d}{2}}\left(1 + (\pi\varepsilon)^{\frac{d}{2}} \exp\left\{\frac{\widehat{a} + R^2}{\varepsilon}\right\}\right). \tag{44}$$

_Step 4._ Now we turn to other expressions. Using the property that a function $t \mapsto \log t$ is $\frac{1}{t_{\min}}$-Lipshitz on interval $[t_{\min}, +\infty)$ we get

$$\log(\exp\{\frac{\widetilde{\psi}(y)}{\varepsilon}\}) - \log(\exp\{\frac{\widetilde{\psi}(y)}{\varepsilon}\} - \delta'') \leq \frac{\delta''}{h_{\min}} \implies$$

$$\log(\exp\{\frac{\widetilde{\psi}(y)}{\varepsilon}\} - \delta'') \geq \frac{\widetilde{\psi}(y)}{\varepsilon} - \frac{\delta''}{h_{\min}}. \tag{45}$$

Similarly, we get that

$$\log(\exp\{\frac{\widetilde{\phi}(y)}{\varepsilon}\}c_\theta(x) - \delta'') \geq \frac{\widetilde{\phi}(y)}{\varepsilon} + \log c_\theta(x) - \frac{\delta''}{e_{\min}}. \tag{46}$$

We use this inequality, monotonicity of logarithm function, and (41), to derive

$$v_{\widetilde{\theta}}(y) \overset{(41)}{>} \exp\{\frac{\widetilde{\psi}(y)}{\varepsilon}\} - \delta'' \implies \log v_{\widetilde{\theta}}(y) > \log(\exp\{\frac{\widetilde{\psi}(y)}{\varepsilon}\} - \delta'') \overset{(45)}{\geq} \frac{\widetilde{\psi}(y)}{\varepsilon} - \frac{\delta''}{h_{\min}} \implies$$

$$-\widetilde{\psi}(y) > -\varepsilon \log v_{\widetilde{\theta}}(y) - \frac{\varepsilon\delta''}{h_{\min}}; \tag{47}$$

$$u_{\widetilde{\omega}}(x) \overset{(43)}{>} \exp\{\frac{\widetilde{\phi}(x)}{\varepsilon}\}c_\theta(x) - \delta'' \implies \log u_{\widetilde{\omega}}(x) > \log(\exp\{\frac{\widetilde{\phi}(x)}{\varepsilon}\}c_\theta(x) - \delta'') \overset{(46)}{\geq}$$

$$\frac{\widetilde{\phi}(y)}{\varepsilon} + \log c_\theta(x) - \frac{\delta''}{e_{\min}} \implies -\widetilde{\phi}(y) > -\varepsilon \log \frac{u_{\widetilde{\omega}}(x)}{c_\theta(x)} - \frac{\varepsilon\delta''}{e_{\min}}. \tag{48}$$

Recall that $\overline{f_1}$ and $\overline{f_2}$ are non-decreasing functions. Moreover, they are Lipshitz with the constants $L_1$, $L_2$ respectively. Thus, we get

$$\overline{f_2}(-\widetilde{\psi}(y)) \geq \overline{f_2}(-\varepsilon \log v_{\widetilde{\theta}}(y) - \frac{\varepsilon\delta''}{h_{\min}}) = \overline{f_2}(-\varepsilon(\log v_\theta(y) + \frac{\|y\|^2}{2\varepsilon}) - \frac{\varepsilon\delta''}{h_{\min}}) =$$

$$\overline{f_2}(-\varepsilon \log v_\theta(y) - \frac{\|y\|^2}{2} - \frac{\varepsilon\delta''}{h_{\min}}) \geq \overline{f_2}(-\varepsilon \log v_\theta(y) - \frac{\|y\|^2}{2}) - L_2 \cdot \frac{\varepsilon\delta''}{h_{\min}}; \tag{49}$$

$$\overline{f_1}(-\widetilde{\phi}(y)) \geq \overline{f_1}(-\varepsilon \log \frac{u_{\widetilde{\omega}}(x)}{c_\theta(x)} - \frac{\varepsilon\delta''}{e_{\min}}) = \overline{f_1}(-\varepsilon \log u_{\widetilde{\omega}}(x) + \varepsilon \log c_\theta(x) - \frac{\varepsilon\delta''}{e_{\min}}) =$$

$$\overline{f_1}(-\varepsilon \log u_\omega(x) - \frac{\|x\|^2}{2} + \varepsilon \log c_\theta(x) - \frac{\varepsilon\delta''}{e_{\min}}) = \overline{f_1}(-\varepsilon \log \frac{u_\omega(x)}{c_\theta(x)} - \frac{\|x\|^2}{2} - \frac{\varepsilon\delta''}{e_{\min}}) \geq$$

$$\overline{f_1}(-\varepsilon \log \frac{u_\omega(x)}{c_\theta(x)} - \frac{\|x\|^2}{2}) - L_1 \cdot \frac{\varepsilon\delta''}{e_{\min}}. \tag{50}$$

Integrating these inequalities over all $x$ and $y$ in supports of $p$ and $q$ respectively, we get

$$\int_{\mathbb{R}^d} \overline{f}_2(-\widetilde{\psi}(y))q(y)dy \geq \int_{\mathbb{R}^d} \overline{f}_2(-\varepsilon \log v_\theta(y) - \frac{\|y\|^2}{2})q(y)dy - L_2 \cdot \frac{\varepsilon\delta''}{h_{\min}}; \tag{51}$$

$$\int_{\mathbb{R}^d} \overline{f}_1(-\widetilde{\phi}(x))p(x)dx \geq \int_{\mathbb{R}^d} \overline{f}_1(-\varepsilon \log \frac{u_\omega(x)}{c_\theta(x)} - \frac{\|x\|^2}{2})p(x)dx - L_1 \cdot \frac{\varepsilon\delta''}{e_{\min}}. \tag{52}$$

Finally, combining (44) and (52), we get

$$\mathcal{L}(\theta, \omega) = \mathcal{J}\Big( \underbrace{\varepsilon \log \frac{u_\omega(x)}{c_\theta(x)} + \frac{\|x\|^2}{2}}_{\phi_{\theta,\omega} \overset{\text{def}}{=}}, \underbrace{\varepsilon \log v_\theta(y) + \frac{\|y\|^2}{2}}_{\psi_\theta \overset{\text{def}}{=}} \Big) =$$

$$\varepsilon \|u_\omega\|_1 + \int_{\mathbb{R}^d} \overline{f}_1(-\varepsilon \log \frac{u_\omega(x)}{c_\theta(x)} - \frac{\|x\|^2}{2})p(x)dx + \int_{\mathbb{R}^d} \overline{f}_2(-\varepsilon \log v_\theta(y) - \frac{\|y\|^2}{2})q(y)dy <$$

$$\varepsilon \int_{\mathbb{R}^d} \int_{\mathbb{R}^d} \exp\{\frac{1}{\varepsilon}(\widetilde{\phi}(x) + \widetilde{\psi}(y) - \frac{\|x-y\|^2}{2})\}dxdy +$$

$$\int_{\mathbb{R}^d} \overline{f}_1(-\widetilde{\phi}(x))p(x)dx + \int_{\mathbb{R}^d} \overline{f}_2(-\widetilde{\psi}(y))q(y)dy +$$

$$L_1 \cdot \frac{\varepsilon\delta''}{e_{\min}} + L_2 \cdot \frac{\varepsilon\delta''}{h_{\min}} + \varepsilon\delta''(2\pi\varepsilon)^{\frac{d}{2}}\Big(1 + (\pi\varepsilon)^{\frac{d}{2}}\exp\{\frac{\widehat{a} + R^2}{\varepsilon}\}\Big) <$$

$$\mathcal{L}^* + \delta' + \delta'' \left[ L_1 \cdot \frac{\varepsilon}{e_{\min}} + L_2 \cdot \frac{\varepsilon}{h_{\min}} + \varepsilon (2\pi\varepsilon)^{\frac{d}{2}} \left( 1 + (\pi\varepsilon)^{\frac{d}{2}} \exp\left\{ \frac{\widehat{a} + R^2}{\varepsilon} \right\} \right) \right] \leq$$

$$\mathcal{L}^* + \frac{\delta\varepsilon}{2} + \frac{\delta\varepsilon}{2} \implies$$

$$D_{\mathrm{KL}}\left( \gamma_{\theta,\omega} \| \gamma^* \right) \leq \varepsilon^{-1} (\mathcal{L}(\theta,\omega) - \mathcal{L}^*) < \delta$$

which completes the proof. $\qquad\square$

*Remark.* In fact, the assumption of the Lishitzness of $\overline{f_1}(x), \overline{f_2}(x)$ can be omitted. Indeed, under the "*everything is compact*" assumptions of Proposition 4.2 and Theorem 4.3, inputs to $\overline{f_1}(\cdot), \overline{f_2}(\cdot)$ also always belong to certain compact sets. The convex functions are known to be Lipshitz on compact subsets of $\mathbb{R}$, see [31, Chapter 3, §18], and we actually do not need the Lipschitzness on the entire $\mathbb{R}$.

## B  Experiments Details

### B.1  General Details

To minimize our objective (14), we parametrize $\alpha_k, r_k, S_k$ of $v_\theta$ and $\beta_l, \mu_l, \Sigma_l$ of $u_\omega$ in (11). Here we follow the standard practices in deep learning and parametrize logarithms $\log \alpha_k$, $\log \beta_l$ instead of directly parameterizing $\alpha_k, \beta_l$. In turn, variables $r_k$, $\mu_l$ are parametrized directly as multi-dimensional vectors. We consider diagonal matrices $S_k$, $\Sigma_l$ and parametrize them via their diagonal values $\log(S_k)_{i,i}$ and $\log(\Sigma_l)_{i,i}$ respectively. We initialize the parameters following the scheme in [41]. In all our experiments, we use the Adam optimizer.

### B.2  Details of the Experiment with Gaussian Mixtures

We use $K = L = 5$, $\varepsilon = 0.05$, $lr = 3e-4$ and batchsize 128. We do $2 \cdot 10^4$ gradient steps. For the LightSB algorithm, we use the parameters presented by the authors in the official repository.

### B.3  Details of the Image Translation Experiment

We use the code and decoder model from

<p align="center">https://github.com/podgorskiy/ALAE</p>

We download the data and neural network extracted attributes for the FFHQ dataset from

<p align="center">https://github.com/ngushchin/LightSB/</p>

In the *Adult* class we include the images with the attribute $Age \geq 44$; in the *Young* class - with the $Age \in [16, 44]$. We excluded the images with faces of children to increase the accuracy of classification per *gender* attribute. For the experiments with our solver, we use weighted $D_{\mathrm{KL}}$ divergence with parameters $\tau$ specified in Appendix C, and set $K = L = 10$, $\varepsilon = 0.05$, $lr = 1$, and batch size to 128. We do $5 \cdot 10^3$ gradient steps using Adam optimizer [35] and `MultiStepLR` scheduler with parameter $\gamma = 0.1$ and milestones$= [500, 1000]$. For testing [41, LightSB] solver, we use the official code (see the link above) and instructions provided by the authors.

**Baselines.** For the OT-FM and UOT-FM methods, we parameterize the vector field $(v_{t,\theta})_{t \in [0,1]}$ for mass transport using a 2-layer feed-forward network with 512 hidden neurons and ReLU activation. An additional sinusoidal embedding[65] was applied for the parameter $t$. The learning rate for the Adam optimizer was set to 1e-4. To obtain an optimal transport plan $\pi^*(x, y)$ discrete OT solvers from the POT [21] package were used. These methods are built on the solutions (plans $\pi^*(x, y)$) of discrete OT problems, to obtain them we use the POT [21] package. Especially for the UOT-FM, we use the `ot.unbalanced.sinkhorn` with the regularization equal to 0.05. We set the number of training and inference time steps equal to 100. To obtain results of UOT-FM for Fig. 3, we run this method for 3K epochs with parameter $reg\_m \in [5e-4, 5e-3, 5e-2, 0.5, 1, 10, 10^2, 10^3, 10^4, 10^5, 10^6]$ and reported the mean values of final metrics for 3 independent launches with different seeds. In Tables 20, 21, 22, for each translation we report the results for one chosen parameter specified in

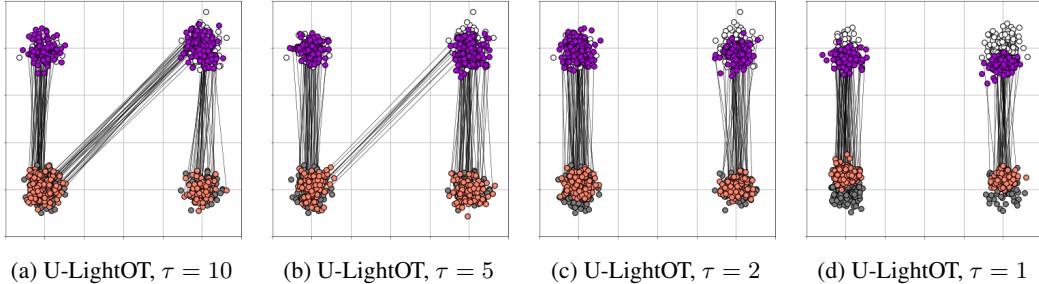

(a) U-LightOT, $\tau = 10$      (b) U-LightOT, $\tau = 5$      (c) U-LightOT, $\tau = 2$      (d) U-LightOT, $\tau = 1$

Figure 5: Conditional plans $\gamma_{\theta,\omega}(y|x)$ learned by our solver with scaled $D_{\chi^2}$ divergences in *Gaussians Mixture* experiment ($\tau \in [1, 2, 5, 10]$).

Appendix D.1. We use the corresponding checkpoints of UOT-FM to visualize its performance in Fig. 4. For [9, UOT-SD], [70, UOT-GAN] we use the official code provided by the authors, see the links:



`https://github.com/Jae-Moo/Unbalanced-Optimal-Transport-Generative-Model`

`https://github.com/uhlerlab/unbalanced_ot`



While both UOT-SB, UOT-GAN methods were not previously applied to the FFHQ dataset, we set up a grid search for the parameters and followed the instructions provided by the authors for parameter settings. Both for UOT-SB, UOT-GAN, we used a 3-layer neural network with 512 hidden neurons, and ReLU activation was used for the generator networks and the potential and discriminator, respectively. Adam optimizer [35] with $lr = 10^{-5}$ and $lr = 10^{-4}$ was used to train the networks in UOT-SB and UOT-GAN, respectively. We train the methods for 10K iterations and set a batch size to 128. For UOT-SD, we used $D_{KL}$ divergence with their unbalancedness parameter $\tau = 0.002$. For other parameters, we used the default values provided by the authors for CIFAR-10 generation tasks. For all baseline models which use entropy regularization, we set $\varepsilon = 0.05$.

## C    Additional Discussion & Experiments with $f$-divergences

**Details about $f$-divergences between positive measures.** In the classic form, $f$-divergences are defined as measures of dissimilarity between two *probability* measures. This definition should be revised when dealing with measures of arbitrary masses. In the paragraph below we show that if the function $f$ is convex, non-negative, and attains zero uniquely at point $\{1\}$ then $D_f(\mu_1\|\mu_2)$ is a valid measure of dissimilarity between two positive measures.

Let $\mu_1, \mu_2 \in \mathcal{M}_{2,+}(\mathbb{R}^{d'})$ be two positive measures. The $f$-divergence satisfies $D_f(\mu_1\|\mu_2) \geq 0$ which is obvious from the non-negativity of $f$. From the definition of $D_f(\mu_1\|\mu_2)$ and the fact that function $f$ attains zero uniquely at a point $\{1\}$, we obtain that $D_f(\mu_1\|\mu_2) = 0$ if and only if $\mu_1(x) = \mu_2(x)$ holds $\mu_2$-everywhere. Actually, $\mu_1(x) = \mu_2(x)$ should hold for all $x$ as $\mu_1$ must be absolutely continuous w.r.t. $\mu_2$ (otherwise $D_f(\mu_1\|\mu_2)$ is assumed to be equal $+\infty$).

**The usage of $D_{\chi^2}$ divergence.** We tested the performance of our solver with scaled $D_{KL}$ divergences in the main text, see §5.1, §5.2. For completeness, here we evaluate our solver with $D_{\chi^2}$ divergence in *Gaussian Mixture* experiment. We use the same experimental setup as in §5.1 and present the qualitative results in Fig. 5. Interestingly, the solver's results differ from those which we obtain for $D_{KL}$ divergence. For $D_{\chi^2}$ divergence, supports of learned plans' marginals constitute only parts of source and target measures' supports when $\tau = 1$. The issue disappears with a slight increase of $\tau$, i.e., for $\tau = 2$. At the same time, a further increase of $\tau$ is useless, since the learned plans fail to deal with class imbalance issue. Thus, parameter $\tau$ should be adjusted heuristically. In the case of $D_{KL}$ divergence, supports coincide for all $\tau$, see Fig. 2. This motivates us to use $D_{KL}$ divergences in our main experiments.

**Parameter $\tau$ in *Gaussian Mixture* experiment.** An ablation study on unbalancedness parameter $\tau$ in *Gaussian Mixture* experiment is conducted in §5.1 and above in this section. For completeness, we also perform the quantitative assessment of our solver with $D_{KL}$ divergence for different unbalancedness parameters $\tau$. We compute the normalized OT cost ($\mathbb{E}_{x\sim p}\mathbb{E}_{y\sim\gamma(y|x)}\frac{(x-y)^2}{2}$) between the source and generated distributions, and the Wasserstein distance between the generated and target

distributions (computed by a discrete OT solver). For completeness, we additionally calculate the metrics for the *balanced* [41, LightSB] approach. The results are presented in Table 4.

| Method | U-LightOT ($\tau = 1$) | U-LightOT ($\tau = 10$) | U-LightOT ($\tau = 50$) | U-LightOT ($\tau = 100$) | LightSB |
|---|---|---|---|---|---|
| OT cost ($\downarrow$) | 2.023 | 2.913 | 3.874 | 3.931 | 3.952 |
| $\mathbb{W}_2$-distance ($\downarrow$) | 2.044 | 1.107 | 0.138 | 0.091 | 0.088 |

Table 4: Normalized OT cost between the input and learned distributions, and Wasserstein distance between the learned and target distributions in *Gaussian mixture* experiment.

*Results.* Recall that the unbalanced nature of our solver leads to two important properties. Firstly, our solver better preserves the properties of the input objects than the balanced approaches − indeed, it allows for preserving object attributes (classes) even in the case of class imbalance. Secondly, due to the relaxed boundary condition for the target distribution, the distribution generated by our solver is naturally less similar to the target distribution than for balanced methods.

The above intuitive reasoning is confirmed by the metrics we obtained. Indeed, as the $\tau$ parameter increases, when our method becomes more and more similar to balanced approaches, the normalized OT cost between the source and generated distributions increases, and the Wasserstein distance between learned and target distributions decreases. LightSB [1] baseline, which is a purely balanced approach, shows the best quality in terms of Wasserstein distance and the worst in terms of OT cost.

**Parameters $\tau$, $\varepsilon$ in image experiments.** The effect of entropy regularization parameter $\varepsilon$ is well studied, see, e.g., [28, 41]. Namely, increasing the parameter $\varepsilon$ stimulates the conditional distributions $\gamma_\theta(y|x)$ to become more dispersed. Still, below we provide an additional quantitative analysis of its influence on the learned translation. Besides, we address the question *how does the parameter $\tau$ influence the performance of our solver in image translation experiments?* To address this question, we learn the translations *Young→ Adult*, *Man→ Woman* on FFHQ dataset varying the parameters $\tau$, $\varepsilon$, see §5.2 for the experimental setup details. We test our solver with scaled $D_{KL}$ divergence training it for 5K iterations. Other hyperparameters are in Appendix B. In Tables 5, 8, 11, 14, we report the accuracy of keeping the attributes of the source images (e.g., gender in *Young→Adult* translation). In Tables 6, 9, 12, 15, we report the accuracy of mapping to the correct target class (e.g., *adult* people in *Young→Adult* translation). In Tables 7, 10, 13, 16, we report FD metrics which is defined as *Frechet distance* between means and covariances of the learned and the target measures. For convenience, we additionally illustrate the results of ablation studies on Fig. 6.

| $\varepsilon$ \ $\tau$ | 10 | 20 | 50 | $10^2$ | **250** | $10^3$ | $10^6$ |
|---|---|---|---|---|---|---|---|
| 0.01 | 96.37±0.07 | 96.12±0.25 | 93.44±0.22 | 90.26±0.18 | 85.15±0.82 | 81.79±0.98 | 79.27±1.11 |
| 0.05 | 96.03±0.08 | 95.55±0.13 | 93.19±0.55 | 88.93±0.94 | 84.49±1.57 | 80.59±0.55 | 78.42±0.73 |
| 0.1 | 95.20±0.21 | 94.92±0.24 | 92.30±0.25 | 87.99±0.83 | 82.91±1.01 | 78.76±0.50 | 78.25±0.87 |
| 0.5 | 88.53±0.26 | 87.46±0.41 | 83.41±0.80 | 80.14±0.56 | 76.00±0.84 | 73.36±0.23 | 71.84±0.27 |
| 1.0 | − | 81.26±1.09 | 77.84±0.33 | 74.33±0.54 | 71.01±1.13 | 67.43±0.15 | 66.71±0.13 |

Table 5: Test accuracy ($\uparrow$) of keeping the class in *Young → Adult* translation.

| $\varepsilon$ \ $\tau$ | 10 | 20 | 50 | $10^2$ | **250** | $10^3$ | $10^6$ |
|---|---|---|---|---|---|---|---|
| 0.01 | 38.94±0.91 | 51.92±0.80 | 67.68±1.06 | 75.49±0.30 | 81.34±1.06 | 83.67±0.74 | 85.27±1.35 |
| 0.05 | 40.09±0.16 | 53.19±0.49 | 69.01±0.74 | 77.41±0.67 | 81.78±0.33 | 84.80±0.90 | 85.63±0.76 |
| 0.1 | 44.18±1.50 | 56.74±0.58 | 71.77±0.57 | 78.34±0.91 | 83.70±0.54 | 87.07±0.37 | 88.21±0.23 |
| 0.5 | 50.51±0.34 | 65.61±2.04 | 81.50±1.17 | 87.48±0.06 | 92.21±0.29 | 92.93±0.14 | 93.80±0.55 |
| 1.0 | − | 70.81±2.69 | 83.82±2.45 | 89.56±0.50 | 93.78±0.35 | 95.04±0.20 | 95.50±0.41 |

Table 6: Test accuracy ($\uparrow$) of mapping to the target in *Young → Adult* translation.

| $\varepsilon$ \ $\tau$ | 10 | 20 | 50 | $10^2$ | **250** | $10^3$ | $10^6$ |
|---|---|---|---|---|---|---|---|
| 0.01 | 35.55±1.24 | 26.77±1.59 | 17.39±0.19 | 16.42±1.96 | 12.28±1.16 | 11.46±0.70 | 10.85±0.12 |
| 0.05 | 42.05±3.04 | 31.47±0.03 | 23.90±1.49 | 18.31±0.16 | 17.15±0.48 | 16.27±0.62 | 19.89±5.51 |
| 0.1 | 50.05±2.48 | 40.67±0.73 | 31.15±0.32 | 27.49±0.56 | 25.86±0.88 | 24.57±0.06 | 26.30±0.15 |
| 0.5 | 114.49±1.06 | 104.42±0.33 | 98.21±4.21 | 92.48±0.98 | 89.42±0.10 | 88.74±0.12 | 89.55±1.18 |
| 1.0 | − | 165.39±1.50 | 149.84±0.86 | 142.39±0.18 | 137.40±0.21 | 135.85±0.17 | 135.25±0.17 |

Table 7: Test FD ($\downarrow$) of generated latent codes in *Young → Adult* translation.

*Results* show that increase of $\varepsilon$ negatively influences both accuracy of keeping the attributes of the source images and FD of generated latent codes which is caused by an increased dispersity of $\gamma_\theta(y|x)$.

| ε \ τ | 10 | 20 | 50 | $\mathbf{10^2}$ | $10^3$ | $10^6$ |
|---|---|---|---|---|---|---|
| 0.01 | 96.14±0.08 | 95.96±0.03 | 93.60±0.72 | 89.88±0.74 | 81.49±0.89 | 80.19±2.17 |
| 0.05 | 95.36±0.41 | 95.26±0.24 | 92.77±0.48 | 89.48±0.60 | 81.34±0.35 | 79.99±0.42 |
| 0.1 | 94.73±0.22 | 94.33±0.21 | 92.65±0.52 | 89.12±0.40 | 81.06±0.82 | 77.73±1.21 |
| 0.5 | 88.21±0.82 | 88.43±0.15 | 86.13±0.94 | 83.89±0.67 | 74.92±0.53 | 70.32±1.46 |
| 1.0 | – | 81.67±0.94 | 79.71±1.28 | 77.15±1.65 | 67.80±0.30 | 65.31±0.90 |

Table 8: Test accuracy (↑) of keeping the class in *Adult → Young* translation.

| ε \ τ | 10 | 20 | 50 | $\mathbf{10^2}$ | $10^3$ | $10^6$ |
|---|---|---|---|---|---|---|
| 0.01 | 67.45±0.51 | 76.38±0.36 | 84.86±0.23 | 87.87±0.37 | 91.56±0.40 | 92.29±0.37 |
| 0.05 | 67.21±0.10 | 76.66±1.22 | 84.43±0.40 | 87.79±0.38 | 91.98±0.50 | 92.29±0.71 |
| 0.1 | 65.29±1.54 | 76.58±1.10 | 84.29±0.73 | 88.65±0.64 | 92.33±0.49 | 92.87±0.11 |
| 0.5 | 69.13±3.43 | 76.74±1.39 | 82.33±6.53 | 86.86±2.89 | 94.06±0.62 | 94.91±0.45 |
| 1.0 | – | 80.13±1.89 | 85.87±0.53 | 91.00±0.62 | 95.66±0.41 | 96.37±0.52 |

Table 9: Test accuracy (↑) of mapping to the target in *Adult → Young* translation.

| ε \ τ | 10 | 20 | 50 | $\mathbf{10^2}$ | $10^3$ | $10^6$ |
|---|---|---|---|---|---|---|
| 0.01 | 45.58±7.90 | 31.24±0.62 | 22.44±0.32 | 18.23±0.24 | 17.30±2.99 | 24.29±12.53 |
| 0.05 | 60.09±13.23 | 38.88±3.33 | 27.85±0.07 | 30.79±8.58 | 24.59±4.48 | 26.44±7.23 |
| 0.1 | 54.45±1.07 | 46.52±0.70 | 41.98±5.22 | 34.02±0.25 | 44.25±9.33 | 40.16±7.51 |
| 0.5 | 128.91±1.09 | 121.00±0.91 | 116.78±10.57 | 111.49±5.02 | 102.51±0.04 | 102.31±0.11 |
| 1.0 | – | 187.33±1.64 | 173.11±1.67 | 171.32±13.12 | 156.25±4.40 | 152.73±0.38 |

Table 10: Test FD (↓) of generated latent codes in *Adult → Young* translation.

| ε \ τ | 10 | 20 | 50 | $\mathbf{10^2}$ | $10^3$ | $10^6$ |
|---|---|---|---|---|---|---|
| 0.01 | 93.75±0.10 | 93.28±0.07 | 92.01±0.16 | 90.85±0.12 | 88.70±0.36 | 87.62±0.34 |
| 0.05 | 93.40±0.17 | 93.20±0.14 | 91.91±0.25 | 90.30±0.39 | 88.05±0.20 | 87.77±0.23 |
| 0.1 | 92.99±0.18 | 92.78±0.15 | 91.24±0.46 | 89.57±0.06 | 87.45±0.62 | 87.21±0.11 |
| 0.5 | 89.96±0.40 | 88.34±0.37 | 87.24±0.26 | 86.75±1.06 | 84.09±0.31 | 83.54±0.41 |
| 1.0 | – | 84.94±0.54 | 83.54±0.41 | 81.80±0.14 | 80.72±0.20 | 79.87±0.47 |

Table 11: Test accuracy (↑) of keeping the class in *Man → Woman* translation.

| ε \ τ | 10 | 20 | 50 | $\mathbf{10^2}$ | $10^3$ | $10^6$ |
|---|---|---|---|---|---|---|
| 0.01 | 61.38±0.29 | 74.57±0.78 | 85.78±0.87 | 90.32±0.70 | 93.79±0.26 | 94.19±0.40 |
| 0.05 | 60.97±1.14 | 74.99±0.32 | 85.01±0.64 | 90.23±0.50 | 93.87±0.25 | 94.44±0.07 |
| 0.1 | 61.36±0.60 | 73.52±0.24 | 86.38±0.40 | 90.38±0.39 | 94.19±0.28 | 94.73±0.36 |
| 0.5 | 65.74±0.87 | 73.61±0.58 | 86.45±0.48 | 89.80±0.66 | 95.14±0.58 | 95.59±0.61 |
| 1.0 | – | 78.18±0.18 | 86.85±0.74 | 91.50±0.74 | 95.63±0.14 | 95.93±0.17 |

Table 12: Test accuracy (↑) of mapping to the target in *Man → Woman* translation.

| ε \ τ | 10 | 20 | 50 | $\mathbf{10^2}$ | $10^3$ | $10^6$ |
|---|---|---|---|---|---|---|
| 0.01 | 54.84±4.12 | 37.87±0.55 | 27.27±3.17 | 22.69±3.48 | 27.73±4.28 | 17.32±1.29 |
| 0.05 | 51.61±2.58 | 51.25±11.78 | 34.28±1.04 | 27.29±2.60 | 30.63±1.67 | 25.77±2.22 |
| 0.1 | 59.45±3.60 | 51.17±3.28 | 43.05±3.68 | 38.02±1.64 | 34.42±0.91 | 37.54±1.79 |
| 0.5 | 132.45±6.07 | 119.82±1.07 | 107.16±1.48 | 107.23±6.76 | 103.24±2.32 | 100.71±1.66 |
| 1.0 | – | 182.26±2.20 | 164.51±1.50 | 156.41±2.04 | 146.73±0.28 | 146.05±0.10 |

Table 13: Test FD (↓) of generated latent codes in *Man → Woman* translation.

| ε \ τ | 10 | 20 | 50 | $10^2$ | $\mathbf{10^3}$ | $10^6$ |
|---|---|---|---|---|---|---|
| 0.01 | 95.70±0.05 | 95.31±0.10 | 93.68±0.09 | 92.46±0.13 | 89.08±0.32 | 88.81±0.05 |
| 0.05 | 95.55±0.08 | 95.05±0.09 | 93.67±0.10 | 92.32±0.27 | 89.66±0.17 | 88.39±0.42 |
| 0.1 | 95.21±0.26 | 94.83±0.19 | 93.14±0.30 | 91.96±0.26 | 89.09±0.53 | 87.38±0.40 |
| 0.5 | 93.02±0.22 | 92.53±0.29 | 91.13±0.01 | 89.73±0.44 | 85.87±1.20 | 85.24±0.41 |
| 1.0 | – | 90.00±0.78 | 88.83±0.73 | 87.46±0.30 | 83.32±1.03 | 83.16±0.55 |

Table 14: Test accuracy (↑) of keeping the class in *Woman → Man* translation.

Interestingly, accuracy of mapping to the correct target class does not have an evident dynamics w.r.t. $\varepsilon$. At the same time, when $\tau$ *increases*, the learned plans provide *worse accuracy* for keeping the input latents' class but *better FD* of generated latent codes and accuracy of mapping to the target

| $\varepsilon$ \ $\tau$ | 10 | 20 | 50 | $10^2$ | $\mathbf{10^3}$ | $10^6$ |
|---|---|---|---|---|---|---|
| 0.01 | 51.54±0.48 | 64.67±1.09 | 77.59±0.39 | 82.52±0.49 | 88.61±0.48 | 88.99±0.18 |
| 0.05 | 49.26±0.72 | 64.09±1.08 | 76.88±0.17 | 83.47±0.52 | 88.59±0.55 | 89.14±0.55 |
| 0.1 | 49.74±0.78 | 63.99±0.82 | 76.96±0.54 | 82.45±0.22 | 89.29±0.55 | 89.14±0.38 |
| 0.5 | 48.64±2.31 | 60.88±0.55 | 77.58±0.28 | 82.10±1.17 | 89.82±0.99 | 90.63±1.06 |
| 1.0 | – | 62.62±0.24 | 74.95±1.17 | 82.42±0.73 | 90.20±0.34 | 90.75±0.37 |

Table 15: Test accuracy ($\uparrow$) of mapping to the target in *Woman $\rightarrow$ Man* translation.

| $\varepsilon$ \ $\tau$ | 10 | 20 | 50 | $10^2$ | $\mathbf{10^3}$ | $10^6$ |
|---|---|---|---|---|---|---|
| 0.01 | 47.00±1.74 | 33.65±0.67 | 23.85±1.15 | 20.83±1.27 | 16.48±0.09 | 18.78±3.44 |
| 0.05 | 52.30±1.29 | 39.79±1.25 | 29.66±1.81 | 27.23±3.45 | 24.68±2.80 | 23.43±1.91 |
| 0.1 | 58.40±0.62 | 48.66±0.73 | 37.55±0.35 | 37.74±2.39 | 31.63±0.42 | 32.83±2.02 |
| 0.5 | 131.17±0.78 | 120.63±0.76 | 108.44±0.60 | 104.85±1.17 | 101.29±0.11 | 102.26±1.58 |
| 1.0 | – | 186.46±0.92 | 169.64±0.63 | 160.52±0.42 | 152.78±0.13 | 152.37±0.09 |

Table 16: Test FD ($\downarrow$) of generated latent codes in *Woman $\rightarrow$ Man* translation.

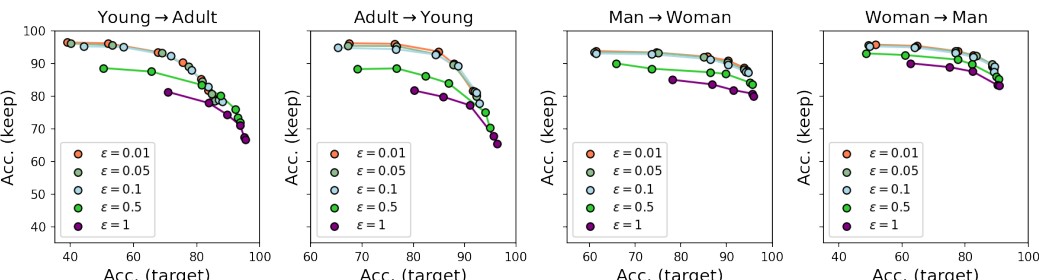

Figure 6: Visualization of ablation studies on parameters $\tau$, $\varepsilon$ in image translation experiment.

class. It is an expected behavior since for bigger $\tau$, the constraints on the marginals of the learned plans become more strict. That is, we enforce the marginals of the learned plans to be closer to source and target measures which allows learning more accurate mappings to target measure but does not allow keeping the source classes in the case of imbalance issues. Interestingly, in *Adult$\rightarrow$Young* translation, FD of learned latents and accuracy of mapping to the target do not change much for $\tau \geq 10^2$ while accuracy of keeping the attributes exhibits a significant drop between $\tau = 10^2$ and $\tau = 10^3$. Thus, we can treat $\tau = 10^2$ as optimal since it provides the best trade-off between the quality of learned translations and their ability to keep the features of input latents. In the case of *Young$\rightarrow$Adult* translation, the values of accuracy and FD exhibit significant differences for considered $\tau$. Thus, we may consider a more detailed scale and choose $\tau = 2.5 \cdot 10^2$ as the optimal one.

It is important to note that our method offers a flexible way to select a domain translation configuration that allows for better preserving the properties of the original objects or generating a distribution closer to the target one. The final optimal configuration selection remains at the discretion of the user. The highlighted values in the Tables are used for comparison with other approaches in §D.1.

*Remark.* FD should be treated as a *relative* measure of similarity between learned and target measures. The results obtained by balanced solver [41, LightSB] (equivalent to ours for big $\tau$) are considered as a gold standard.

**Number of Gaussian components in potentials.** For completeness, we perform an ablation study of our U-LightOT solver with different number on Gaussian components ($K$, $L$) in potentials $v_\theta$ and $u_\omega$, respectively. We run the solver in *Young$\rightarrow$Adult* translation with 5K steps, $\varepsilon = 0.05$ and set $\tau = 250$. The quantitative results (accuracy of keeping the class, accuracy of mapping to the target, FD of generated latents vs target latents) are presented in the Tables below.

The results show that in the considered task, our solver provides good performance even for small number of Gaussian components. This can be explained by the smoothness of the latent representations of data in ALAE autoencoder.

| K\L | 1 | 2 | 4 | 8 | 16 | 32 | 64 | 128 |
|---|---|---|---|---|---|---|---|---|
| 1 | 84.17±0.45 | 84.73±1.27 | 84.25±1.08 | 85.08±0.78 | 84.84±0.92 | 85.82±0.70 | 85.52±0.74 | 83.70±0.48 |
| 2 | 84.99±0.59 | 84.54±0.43 | 84.56±0.39 | 84.18±0.63 | 83.87±2.03 | 83.56±1.37 | 84.66±0.44 | 86.74±0.54 |
| 4 | 84.50±0.86 | 84.81±0.46 | 84.60±0.37 | 84.65±0.77 | 83.75±1.01 | 84.00±1.52 | 83.51±0.90 | 83.99±0.63 |
| 8 | 83.88±1.01 | 84.08±0.81 | 83.71±0.60 | 82.76±1.98 | 84.69±0.38 | 84.30±0.39 | 85.05±1.59 | 83.03±1.02 |
| 16 | 84.65±0.33 | 85.00±1.62 | 84.28±0.78 | 84.76±1.23 | 83.66±0.33 | 85.14±0.49 | 83.77±1.16 | 84.34±0.15 |
| 32 | 83.48±0.40 | 86.02±1.22 | 84.79±0.60 | 84.44±0.42 | 85.24±0.84 | 84.06±1.00 | 84.73±0.51 | 84.54±0.27 |
| 64 | 85.24±0.27 | 85.23±0.59 | 84.12±0.12 | 84.64±0.64 | 84.01±1.95 | 84.10±1.26 | 84.77±1.25 | 83.76±1.44 |
| 128 | 85.21±0.29 | 85.16±0.07 | 84.63±1.15 | 84.64±0.49 | 84.12±0.57 | 84.11±0.76 | 84.22±0.68 | 84.64±0.93 |

Table 17: Test accuracy (↑) of keeping the attributes in *Young→Adult* translation for our U-LightOT solver with different number of Gaussian components in potentials.

| K\L | 1 | 2 | 4 | 8 | 16 | 32 | 64 | 128 |
|---|---|---|---|---|---|---|---|---|
| 1 | 83.35±0.85 | 82.03±0.11 | 82.69±0.82 | 83.34±0.08 | 82.95±0.46 | 82.80±0.58 | 82.49±0.64 | 82.54±0.36 |
| 2 | 83.26±0.98 | 81.80±0.81 | 81.82±0.66 | 82.35±0.92 | 82.77±0.45 | 82.26±0.98 | 82.36±0.37 | 82.84±0.38 |
| 4 | 82.76±0.17 | 81.47±0.29 | 82.32±0.13 | 82.60±0.66 | 82.70±0.64 | 82.48±0.37 | 83.23±1.24 | 82.44±0.68 |
| 8 | 82.35±0.39 | 81.63±0.38 | 82.23±0.52 | 82.19±0.72 | 82.64±0.58 | 82.55±0.18 | 82.85±0.28 | 82.87±0.09 |
| 16 | 82.43±0.61 | 82.12±1.10 | 81.30±0.79 | 82.11±0.93 | 82.56±0.26 | 81.88±0.85 | 83.34±0.57 | 82.90±0.39 |
| 32 | 81.57±0.47 | 81.39±0.62 | 81.76±0.27 | 81.28±0.57 | 82.47±1.30 | 82.06±0.41 | 82.35±0.55 | 82.56±0.89 |
| 64 | 82.01±0.79 | 82.33±0.83 | 81.97±0.61 | 82.01±0.26 | 82.44±0.74 | 81.61±0.75 | 82.06±0.51 | 83.44±0.38 |
| 128 | 82.39±0.92 | 82.18±0.56 | 81.74±0.19 | 82.22±0.74 | 81.58±0.23 | 82.66±0.33 | 83.07±0.30 | 82.85±0.45 |

Table 18: Test accuracy (↑) of mapping to the target in *Young→Adult* translation for our U-LightOT solver with different number of Gaussian components in potentials.

| K\L | 1 | 2 | 4 | 8 | 16 | 32 | 64 | 128 |
|---|---|---|---|---|---|---|---|---|
| 1 | 17.41±0.28 | 17.42±0.36 | 18.10±1.41 | 17.83±0.41 | 17.97±0.69 | 18.42±0.86 | 17.69±0.37 | 18.07±0.61 |
| 2 | 19.07±0.81 | 17.47±0.83 | 16.87±0.25 | 16.93±0.56 | 21.05±1.86 | 17.38±0.69 | 17.22±0.76 | 17.83±0.70 |
| 4 | 17.07±0.53 | 16.66±0.14 | 17.02±0.84 | 18.01±1.79 | 16.71±0.06 | 17.07±0.95 | 16.59±0.41 | 16.42±0.10 |
| 8 | 16.55±0.22 | 23.56±7.59 | 16.37±0.23 | 16.81±0.95 | 17.21±1.01 | 18.37±1.47 | 17.13±0.77 | 16.96±0.86 |
| 16 | 17.58±1.58 | 18.06±1.61 | 18.25±2.46 | 17.69±1.87 | 18.00±0.34 | 17.19±0.71 | 17.62±0.93 | 18.97±2.01 |
| 32 | 17.85±0.10 | 17.15±0.71 | 16.50±0.10 | 21.91±4.05 | 20.04±2.78 | 19.31±3.22 | 18.79±2.03 | 22.56±1.66 |
| 64 | 21.39±1.32 | 21.91±4.23 | 21.00±4.47 | 18.02±1.24 | 21.63±3.14 | 19.56±0.99 | 17.53±0.47 | 20.26±4.46 |
| 128 | 22.09±4.91 | 35.68±21.07 | 33.80±21.67 | 20.46±2.39 | 24.80±5.33 | 22.73±3.03 | 22.09±1.25 | 22.72±7.41 |

Table 19: Test FD (↓) of generated latent codes in *Young→Adult* translation for our U-LightOT solver with different number of Gaussian components in potentials.

# D  Additional Experimental Results

## D.1  Quantitative comparison with other methods in Image-to-Image translation experiment

In this section, we provide additional results of quantitative comparison of our U-LightOT solver and other unbalanced and balanced OT/EOT solvers in image translation experiment. Tables 20, 21, 22, provide values used for plotting Fig. 3 in the main text. The unbalancedness parameters used for our U-LightOT solver and [16, UOT-FM] are specified in th Tables below. For obtaining the result of [9, UOT-SD], we use their unbalancedness parameter $\tau = 0.002$. For other details on methods' parameters used to obtain the results below, see Appendix B. Note that we do not include FID metric for assessing the quality of generated images since we found that it is not a representative metric for assessing the performance of models performing the translation of ALAE latent codes.

| Experiment | OT-FM [16] | [41, LightSB] *or* [30, LightSBM] | [16, UOT-FM] | UOT-SD [9] | UOT-GAN [70] | U-LightOT (**ours**) |
|---|---|---|---|---|---|---|
| *Young→Adult* | 67.71 | 78.16 | 84.46 ($reg\_m = 0.005$) | 45.71 | 73.85 | 84.49 ($\tau = 250$) |
| *Adult→Young* | 53.79 | 80.25 | 76.05 ($reg\_m = 0.005$) | 49.30 | 74.74 | 89.48 ($\tau = 10^2$) |
| *Man→Woman* | 76.05 | 87.82 | 84.42 ($reg\_m = 0.005$) | 75.50 | 84.04 | 90.30 ($\tau = 10^2$) |
| *Woman→Man* | 72.40 | 88.10 | 86.10 ($reg\_m = 0.05$) | 72.03 | 84.56 | 89.66 ($\tau = 10^3$) |

Table 20: Comparison of accuracies of keeping the attributes of the source images.

| Experiment | OT-FM [16] | [41, LightSB] *or* [30, LightSBM] | [16, UOT-FM] | UOT-SD [9] | UOT-GAN [70] | U-LightOT (**ours**) |
|---|---|---|---|---|---|---|
| *Young→Adult* | 93.28 | 85.97 | 74.10 ($reg\_m = 0.005$) | 87.33 | 84.25 | 81.78 ($\tau = 250$) |
| *Adult→Young* | 96.12 | 93.10 | 89.30 ($reg\_m = 0.005$) | 97.39 | 95.88 | 87.79 ($\tau = 10^2$) |
| *Man→Woman* | 93.33 | 94.37 | 92.35 ($reg\_m = 0.005$) | 98.16 | 97.38 | 90.23 ($\tau = 10^2$) |
| *Woman→Man* | 94.27 | 89.66 | 88.53 ($reg\_m = 0.05$) | 94.96 | 92.91 | 88.59 ($\tau = 10^3$) |

Table 21: Comparison of accuracies of mapping to the target.

| Experiment | OT-FM [16] | [41, LightSB] *or* [30, LightSBM] | [16, UOT-FM] | UOT-SD [9] | UOT-GAN [70] | U-LightOT (**ours**) |
|---|---|---|---|---|---|---|
| *Young→Adult* | 11.93 | 15.50 | 11.57 ($reg\_m = 0.005$) | 13.28 | 11.23 | 17.15 ($\tau = 250$) |
| *Adult→Young* | 14.10 | 21.41 | 17.00 ($reg\_m = 0.005$) | 18.44 | 14.94 | 30.79 ($\tau = 10^2$) |
| *Man→Woman* | 16.20 | 20.91 | 10.31 ($reg\_m = 0.005$) | 16.13 | 22.41 | 27.29 ($\tau = 10^2$) |
| *Woman→Man* | 11.42 | 30.87 | 6.99 ($reg\_m = 0.05$) | 13.23 | 10.55 | 24.68 ($\tau = 10^3$) |

Table 22: Comparison of FD between the generated and learned latents.

### D.2 Outlier robustness property of U-LightOT solver

To show the outlier robustness property of our solver, we conduct the experiment on Gaussian Mixtures with added outliers and visualize the results in Fig. 7. The setup of the experiment, in general, follows the *Gaussian mixtures* experiment setup described in section 5.2 of our paper. The difference consists in outliers (small gaussians) added to the input and output measures.

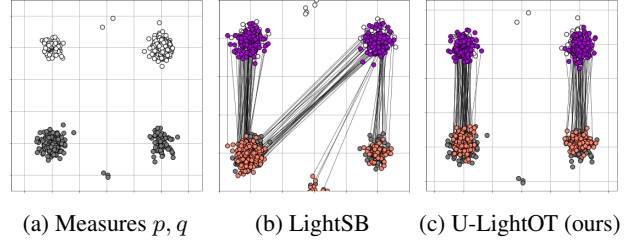

    (a) Measures $p, q$      (b) LightSB      (c) U-LightOT (ours)

Figure 7: Conditional plans $\gamma_{\theta,\omega}(y|x)$ learned by our solver ($\tau = 1$) and LightSB in *Gaussian Mixtures with otliers* experiment.

*The results* show that our U-LightOT solver successfully eliminates the outliers and manages to simultaneously handle the class imbalance issue. At the same time, the balanced LightSB [41] solver fails to deal with either of these problems.

## E   Limitations and Broader Impact

**Limitations.** One limitation of our solver is the usage of the Gaussian Mixture parametrization which might restrict the scalability of our solver. This points to the necessity for developing ways to optimize objective (4) with more general parametrization, e.g., neural networks.

**Broader impact.** Our work aims to advance the field of Machine Learning. There are many potential societal consequences of our work, none of which we feel must be specifically highlighted here.

