# OpenReview forum: "Light Unbalanced Optimal Transport"
_NeurIPS.cc/2024/Conference — NeurIPS 2024 poster_

### Official Review · Reviewer_WAcu · 2024-07-10

**Soundness:** 2
**Presentation:** 3
**Contribution:** 3
**Rating:** 5
**Confidence:** 3

**Summary:**

This paper proposes U-LightOT, a lightweight solver for the Unbalanced Entropic Optimal Transport (UEOT) problem. This method uses a Gaussian Mixture approximation for the potential $v_{\theta}(y)$ and measure $u_{w}(x)$. This paper proves that under this approximation, the KL divergence to the ground truth UEOT plan has a tractable form. U-LightOT is evaluated on Gaussian Mixture and Unpaired Image-to-Image Translation tasks.

**Strengths:**

1. This paper provides a theoretical analysis of the generalization bounds and the universal approximation property for the Gaussian mixture parametrization.
2. The proposed method is a lightweight solver for the UEOT problem, which requires several minutes of CPU training for the experiment in Sec 5.
3. This paper is easy to follow.

**Weaknesses:**

1. The optimization objective and Gaussian Mixture approximation in Sec 4 are similar to [1].

2. While this paper provides the universal approximation property for Gaussian Mixure approximation, I have concerns about whether this Gaussian Mixture parametrization can achieve decent results for more complex distributions, such as in generative modeling within the data space on CIFAR-10.

[1] Korotin, Alexander, Nikita Gushchin, and Evgeny Burnaev. "Light schr\" odinger bridge." ICLR 2024.

**Questions:**

1. In the Unpaired Image-to-Image Translation task, Table 2 only presents the accuracy of keeping the attributes of the source images. However, since the goal of this task is semantic translation, the accuracy of the target semantic is also required. For example, in the Young-to-Adult task, the accuracy of whether the generated image is indeed an adult image. Could you provide this target semantic accuracy results?
2. In the Appendix, Tables 5 and 7 show the Frechet distance (FD) between the learned and target measures. I believe this FD metric evaluates whether the semantic translation is successful, at the marginal level. Generally, increasing τ\tauτ decreases (improves) FD metrics in Table 5 and 7. Could you clarify how the optimal $\tau$ is selected? I am curious because when $\tau$ is overly large, U-LightOT achieves worse accuracy compared to other models in Table 2.
3. Could you present the FD results (Tables 5 and 7) for the other models?
4. For the optimal $\tau=500$ in the Man-to-Woman task, Table 6 shows an accuracy of 83.85 at best, while Table shows an accuracy of 92.85. Could you clarify which result is correct?

**Limitations:**

The authors addressed the limitations and broader impact of their work.

---

> ### Author Rebuttal · Authors · 2024-08-07
>
> Thank you for your thorough feedback. Please find the answers to your questions below.
>
> **(1) The optimization objective and Gaussian Mixture approximation in Sec 4 are similar to [1].**
>
> Our solver can be considered as the generalization of the one from [LightSB, 1] in the sense that it subsumes LightSB for the specific choice of $f$-divergences. However, this generalization is not straightforward or direct since our objective is built on the completely different principles:
>
> 1. Our solver is derived from minimizing $D_{\text{KL}}$ divergence (defined as a discrepancy between positive measures) between ground truth plan $\gamma^*$ and its approximation $\gamma_{\theta,\omega}$. This definition of divergence notably differs from the ordinary definition of $D_{\text{KL}}$ for probability measures used in [1].
>
> 2. We parametrize the entire plan using Gaussian mixtures while in [1] it is done only for conditional plans. It is an important difference, since in an unbalanced case the marginals of optimal plan do not coincide with source and target measures. Our parametrization allows sampling from the left marginal of UEOT plan and identifying potential outliers in the source measure.
>
> **(2) While this paper provides the universal approximation property for Gaussian Mixure approximation, I have concerns about whether this Gaussian Mixture parametrization can achieve decent results for more complex distributions, such as in generative modeling within the data space on CIFAR-10.**
>
> In general, methods based on Gaussian parametrization are usually not appropriate for tasks with complex data, e.g., images. We mention this limitation in our paper  (lines 256-257,702-707). Still, our aim was to develop a lightweight unbalanced solver which can serve as a simple and easy-to-use baseline in moderate-dimensional tasks. As expected, we get this ease in exchange for the rich parametrization required for large-dimensional tasks and vice versa.
>
> **(3) In the Unpaired Image-to-Image Translation task, Table 2 only presents the accuracy of keeping the attributes of the source images. However, since the goal of this task is semantic translation, the accuracy of the target semantic is also required. For example, in the Young-to-Adult task, the accuracy of whether the generated image is indeed an adult image. Could you provide this target semantic accuracy results?**
>
> We are working on this and aim to add the results soon.
>
> **(4) In the Appendix, Tables 5 and 7 show the Frechet distance (FD) between the learned and target measures. I believe this FD metric evaluates whether the semantic translation is successful, at the marginal level. Generally, increasing $\tau$ decreases (improves) FD metrics in Table 5 and 7. Could you clarify how the optimal tau is selected? I am curious because when tau is overly large, U-LightOT achieves worse accuracy compared to other models in Table 2.**
>
> In Appendix C (Tables 5, 7), we perform an ablation study of our method in order to show that it offers a flexible way to select a domain translation configuration (unbalancedness parameter $\tau$) that either allows for very good level of preserving the properties of the input objects or generation of a distribution which is a very good approximation of the target distribution. In that section, we highlighted the parameter which is optimal in the sense that it provides the best tradeoff between the closeness of the learned translations and target ones (*Pareto-optimal*), and the ability of the learned latents to keep the features of input latents. However, the final selection of the optimal configuration remains at the discretion of the user.
>
>
> **(5) Could you present the FD results (Tables 5 and 7) for the other models?**
>
> We will add the results soon.
>
> **(6) For the optimal tau=500 in the Man-to-Woman task, Table 6 shows an accuracy of 83.85 at best, while Table shows an accuracy of 92.85. Could you clarify which result is correct?**
>
> The Tables provide results for different number of training steps and parameters $\tau$. Table 2 shows the accuracy for $\tau=100$ and 5K steps of the algorithm which is specified in Appendix B.3. Table 6 presents the results for only 3K steps and different parameters $\tau$. (Note that the value for $\tau=100$ in the Table 6 (88.59 $\pm$ 0.40) is close to 92.85 - up to the difference in number of training steps.)
>
> **References.**
>
> [1] Korotin, Alexander, Nikita Gushchin, and Evgeny Burnaev. "Light schr" odinger bridge." ICLR 2024.

---

> > ### Author Response · Authors · 2024-08-10
> > **Further clarifications**
> >
> > **In the Unpaired Image-to-Image Translation task, Table 2 only presents the accuracy of keeping the attributes of the source images. However, since the goal of this task is semantic translation, the accuracy of the target semantic is also required. For example, in the Young-to-Adult task, the accuracy of whether the generated image is indeed an adult image. Could you provide this target semantic accuracy results? [...] Could you present the FD results (Tables 5 and 7) for the other models?**
> >
> > As per your request, we provide the Accuracy (of mapping to the target) and FD (between learned and target latents) results for our solver and its unbalanced competitors (in Young$\rightarrow$Adult translation) in the Table below. For completeness, we also include the results for balanced LightSB [1] solver.
> >
> > |                                           | Choi et al. [3]  | Yang et al. [4]  | UOT-FM [2]   | LightSB [1] | ULight-OT (ours, $\tau=100$) |
> > |-------------------------------------------|-------|-------|-------|---------|-------|
> > | Accuracy (mapping to the target)          | 85.36 | 80.32 | 83.27 | 88.14   | 81.14 |
> > | FD (between generated and target latents) | 13.24 | 11.50 | 10.27 | 24.66   | 27.72 |
> >
> > The results show that *balanced* LightSB solver outperforms other methods according to the target accuracy results. Note that FD metrics is based on the first and second moments of distributions, therefore, there is a chance that it can provide imprecise results (as it, possibly, happens for the case of LightSB). Our method provides the target accuracy results on par with UOT-FM model (which is the second-best model according to the accuracy of keeping the class, see Table 2). Other unbalanced solvers (Yang et al., Choi et al.) provide better accuracies of mapping to the target and FD results, but are slightly worse in keeping the attributes (classes) of the source latents.  Besides, our solver is simpler and and faster than its competitors, especially, those from (Yang et al., Choi et al.) which are based on adversarial learning, see the speed-up comparison in our answer to the reviewer bWBu (https://openreview.net/forum?id=co8KZws1YK&noteId=akup4rEAWd).
> >
> > [1] Korotin et al. "Light schrodinger bridge", ICLR 2024.
> >
> > [2] L. Eyring et al. Unbalancedness in neural monge maps improves unpaired domain translation. ICLR, 2024
> >
> > [3] J. Choi et al. Generative modeling through the semi-dual formulation of unbalanced optimal transport. NeurIPS, 2023.
> >
> > [4] K. D. Yang et al. Scalable unbalanced optimal transport using generative adversarial networks. ICLR, 2018.

---

> > > ### Comment · Reviewer_WAcu · 2024-08-11
> > >
> > > I appreciate the author for their clarifications and additional experiments. These have been helpful in addressing my concerns. Hence, I will raise my rating to 5.

---

### Official Review · Reviewer_bWBu · 2024-07-11

**Soundness:** 4
**Presentation:** 4
**Contribution:** 3
**Rating:** 6
**Confidence:** 4

**Summary:**

The proposal focuses on developing a fast solver for the unbalanced entropy-regularized optimal (EOT) transport between continuous Radon measures. The authors utilize the dual formulation of unbalanced EOT and use the relationship between the optimal potentials (i.e., the dual variables) and the primal transport plan. They then consider a parameterization of the transport plan and plan to minimize the KL divergence between this parameterized transport plan and the optimal one. Given that the optimal transport plan is unknown, the authors first use the relationship between the primal and dual solutions to reparameterize the dual variables and then derive a tight upper bound for the KL between the optimal plan and the parameterized one, which they propose to minimize. To deal with the normalization terms in their upper bound, the authors use a similar framework to that of Gushchin et al. [29] and assume the reparameterized dual variables are unnormalized Gaussian mixtures; this assumption enables analytic solutions to the otherwise difficult-to-calculate terms in the upper bound. Lastly, the authors provide a generalization error bound for their proposed framework. The paper provides two small-scale numerical examples to demonstrate their solver's efficiency: 1) two-dimensional Gaussian mixtures and 2) unpaired-image-to-image translation in the embedding space of an autoencoder, specifically ALAE, on an unbalanced subset of FFHQ dataset for Adult, Young, Man, Woman face.

**Strengths:**

+ The paper is very well written and straightforward to follow.
+ The clever parameterizations used in this paper (while they appear in some prior work), provide a unique approach for solving the UEOT problem between continuous measures.
+ The provided generalization error bounds (while straightforward to derive), are important and certainly add value to the paper.
+ The method is easy to implement and fast to train. Quick convergence on the CPU is a notable achievement unlocked by this work.

**Weaknesses:**

- One major weakness is that the paper does not discuss how $K$ and $L$, i.e., the number of Gaussians in the mixtures, affect the results. The generalization error bound mentions that $K$ and $L$ will appear as constants in the error bound, but the practical implications of the choice of $K$ and $L$ are missing from the paper.

- Experiments are relatively modest: 2 experiments in low dimensions. It would be beneficial to have an experiment on robustness to outliers, as this is included in the main claim.

- The paper claims to have a fast solver but lacks a detailed speed comparison for either experiment. It would be great to have a wall-clock comparison of competing methods.

- The Gaussian mixture assumption limits the method's applicability to only low-dimensional problems, and it is not clear whether this limitation can be overcome.

**Questions:**

* How does the performance change as a function of $K$ (assuming $L=K$)?

**Limitations:**

Limitations are provided in the appendix.

---

> ### Author Rebuttal · Authors · 2024-08-07
>
> Thank you for your thorough feedback. Please find the answers to your questions below.
>
> **(1) The paper does not discuss how K and L, i.e., the number of Gaussians in the mixtures, affect the results. The generalization error bound mentions that K and L will appear as constants in the error bound, but the practical implications of the choice of K and L are missing from the paper. [...] How does the performance change as a function of
>  K(assuming K=L)?**
>
> To address your question, we perform additional experiments (both on Gaussians and in the latent space of ALAE autoencoder) with our solver varying the number of Gaussian modes ($K$, $L$) in potentials.
>
> The setup of this experiment follows the setup introduced in our Section 5.1. We test our solver with diverse number of potentials $K\in\{1,3,5\}$ and  $L\in\{1,2,3,4,5\}$. The results are visualized in Fig. 1 of the **attached PDF file**. It can be seen that for insufficient number of modes in potentials, the solver exhibits issues with convergence and do not correctly solve the task.
>
> **(2) It would be beneficial to have an experiment on robustness to outliers, as this is included in the main claim.**
>
> Thank you for this valuable suggestion. We conduct the experiment on Gaussian Mixtures with added outliers and visualize the results in Fig. 2 of the **attached PDF file**. The setup of the experiment, in general, follows the *Gaussian mixtures* experiment setup described in section 5.2 of our paper. The difference consists in outliers (small gaussians) added to the input and output measures. The results show that our U-LightOT solver successfully eliminates the outliers and manages to simultaneously handle the class imbalance issue. At the same time, the balanced LightSB [4] solver fails to deal with either of these problems.
>
> **(3) [...] speed comparison for either experiment. It would be great to have a wall-clock comparison of competing methods.**
>
> Thank you for the suggested idea. We compared the running time of our algorithm and unbalanced competitors on the image translation task (Adult$\rightarrow$Young), its setup is described in section 5.2 of our paper. The results for all of the methods (wall-clock times for 10k updates) are presented in the Table below. We omit the results for other variants of translations since they are quite similar to the results in the Table.
>
> As you can see, our proposed solver outperforms its competitors (unbalanced methods) in terms of convergence time.
>
> |      | ULight-OT | UOT-FM [1] | Yang et al. [2] | Choi et al. [3] |
> |---|---|---|---|---|
> | Time | **02:38**     | *03:21*   | 16:30      | 18:11 |
>
> **(4) The Gaussian mixture assumption limits the method's applicability to only low-dimensional problems, and it is not clear whether this limitation can be overcome.**
>
> In general, methods based on Gaussian parametrization are usually not appropriate for tasks with complex data, e.g., images. We mention this limitation in our paper  (lines 256-257,702-707). Still, our aim was to develop a lightweight unbalanced solver which can serve as a simple and easy-to-use baseline in moderate-dimensional tasks. As expected, we get this ease in exchange for the rich parametrization required for large-dimensional tasks and vice versa.
>
> **Concluding remarks**. Please respond to our post to let us know if the clarifications above suitably address your concerns about our work. We are happy to address any remaining points during the discussion phase; if the responses above are sufficient, we kindly ask that you consider raising your score.
>
> **References.**
>
> [1] L. Eyring et al. Unbalancedness in neural monge maps improves unpaired domain translation. ICLR, 2024
>
> [2] J. Choi et al. Generative modeling through the semi-dual formulation of unbalanced optimal transport. arXiv preprint arXiv:2305.14777, 2023.
>
> [3] K. D. Yang et al. Scalable unbalanced optimal transport using generative adversarial networks. ICLR, 2018.
>
> [4] Korotin et al. "Light schrodinger bridge", ICLR 2024.

---

> > ### Comment · Reviewer_bWBu · 2024-08-10
> > **Response to rebuttal**
> >
> > I appreciate the authors' extensive responses and clarifications.
> >
> > The experiment on varying $K$ and $L$ is particularly insightful, as it demonstrates that even in a simple toy problem, the method's performance is highly sensitive to the appropriate selection of these hyperparameters. I believe the paper would benefit from reporting the method's results on large-scale experiments across a range of $K$ and $L$ values, perhaps in the supplementary material.
> >
> > I also appreciate the wall-clock performance data provided by the authors. A more rigorous analysis of the wall-clock time, considering different sample sizes and varying $K$ and $L$ values on toy datasets, could further enhance the paper’s practical value to the community.
> >
> > Overall, I find this paper well-written, easy to follow, novel, and of potential interest to the community. The strengths of the paper outweigh the weaknesses, and I am increasing my score to Weak Accept.

---

### Official Review · Reviewer_vYvs · 2024-07-12

**Soundness:** 3
**Presentation:** 2
**Contribution:** 3
**Rating:** 6
**Confidence:** 2

**Summary:**

This work focuses on the largely computationally intractable efforts in unbalanced OT dual form where neural networks are used as a proxy (used as potentials) in order to approximate Wasserstein distances. In this work, the authors set out to significantly reduce this optimization procedure by decomposing the join optimal solution into conditionals which allows for both easier inference and a reduction in the number of parameters required. Experimental results are then carried out to show the success of this method beyond the improved efficiency.

**Strengths:**

[+] The reduction in efficiency seems to be quite strong and effective way to reduce the overall parameters required to approximate OT distances
[+] The theoretical results are sound and well motivated.
[+] A generalization bound is also presented, attesting to the soundness one achieves with this light variation.

**Weaknesses:**

[-] Appears to be specific only to the case of having KL divergence penalties for the mass constraints.
[-] Paper can appear a bit difficult and dense to read.

**Questions:**

(1) The reduction you get appears to have (perhaps even superficially) some relationship to the way WAE decomposes the coupling into conditionals. More coincidentally, WAE also uses conditional Gaussians to parametrize the encoder distribution, although for different purposes. Do you have any comments if there is any deeper link here?

(2) Do you have any intuition if one were to use other penalties beyond KL to enforce the mass constraint?

**Limitations:**

Yes

---

> ### Author Rebuttal · Authors · 2024-08-07
>
> Thank you for your thorough feedback. Please find the answers to your questions below.
>
> **(1) Appears to be specific only to the case of having $D_{\text{KL}}$ divergence penalties for the mass constraints. [...] Do you have any intuition if one were to use other penalties beyond $D_{\text{KL}}$ to enforce the mass constraint?**
>
> Our solver admits different divergences except for the $D_{\text{KL}}$ one. From the theoretical point of view, we describe the set of admissible divergences in our Appendix C (lines 549-672). Besides, we provide a numerical example illustrating the performance of our solver with $\mathcal{D}_{\chi^2}$ divergence, see Fig. 3 and description in lines 559-677.
>
> **(2) Paper can appear a bit difficult and dense to read.**
>
> We are upset that you found our work difficult to read. We will try to improve this aspect if you indicate in more detail which points were difficult to understand.
>
> **(3) The reduction you get appears to have (perhaps even superficially) some relationship to the way WAE decomposes the coupling into conditionals. More coincidentally, WAE also uses conditional Gaussians to parametrize the encoder distribution, although for different purposes. Do you have any comments if there is any deeper link here?**
>
> Thanks for asking. We think that there is no direct link. Indeed, in WAE, the encoder for each input $x$ outputs some Gaussian, while in our case, all the conditional Gaussians (more precisely, Gaussian mixtures) are tight together. This means that given one conditional distribution $\gamma_{\theta}(y|x=x_0)$, one can immediately express all the other $\gamma_{\theta}(y|x=x_{\text{other}})$. In fact, the densities of all these conditional distributions are parameterized by a single scalar-valued function $v$; see eq. (8) in our paper. This is achieved because of the properties of the entropic optimal transport solutions which we exploited to construct our algorithm.
>
> **Concluding remarks**. Please respond to our post to let us know if the clarifications above suitably address your concerns about our work. We are happy to address any remaining points during the discussion phase; if the responses above are sufficient, we kindly ask that you consider raising your score.

---

> > ### Comment · Reviewer_vYvs · 2024-08-11
> >
> > Thank you for responding to my questions, I don't have any concerns after reading the response.

---

### Official Review · Reviewer_t9a3 · 2024-07-12

**Soundness:** 3
**Presentation:** 3
**Contribution:** 2
**Rating:** 6
**Confidence:** 3

**Summary:**

The paper presents a novel approach to solving the continuous Unbalanced Entropic Optimal Transport (UEOT) problem. The authors introduce a lightweight, theoretically-justified solver that addresses the challenges of sensitivity to outliers and class imbalance in traditional Entropic Optimal Transport (EOT). The proposed method features a non-minimax optimization objective and employs Gaussian mixture parametrization for UEOT plans, resulting in a fast, simple, and effective solver. The authors provide theoretical guarantees for their solver's performance and apply it to simulated and image data.

**Strengths:**

- The paper is well-written and easy to follow
- The paper describes well related literature and clearly motivates the approach / why there is a need for this solver
- The paper introduces a novel way to solve UEOT problems using Gaussian mixtures, even if the approach was previously used for balanced EOT problems as mentioned by the authors.
- The authors thoroughly study generalization bounds.
- The authors consider a wide range of competing methods.

**Weaknesses:**

- While the authors provide generalisation bounds, it would be helpful to assess the performance of the method on the UEOT plan between Gaussian distributions, see Janati et al. ,2020
- As mentioned by the authors, the Gaussian mixture approach is likely to work only in low dimensions. It would be interesting to see when it fails, e.g. using the benchmark above.

**Questions:**

- The authors state that for OT-FM and UOT-FM in the FFHQ dataset, they use a 2-layer feed-forward network with 512 hidden neurons and ReLU activation. Where does this parameterization come from? It seems to be relatively small for a flow matching architecture on images, and does not seem to be the architecture used in the original papers.
- Why are FID scores not reported for the image translation tasks?

**Limitations:**

The authors have considered the limitations and potential negative societal impact.

---

> ### Author Rebuttal · Authors · 2024-08-07
>
> Thank you for your thorough feedback. Please find the answers to your questions below.
>
> **(1) Performance of the method on the UEOT plan between Gaussian distributions, see Janati et al.,2020 [5]**
>
> Thank you for your suggestion. Unfortunately, a comparison of our method's solutions with the analytical solutions proposed in [5] is not relevant, since this paper consideres a different setup of the UEOT problem. Namely, this paper derives solutions for the UEOT problem (between Guassian measures) with $D_{\text{KL}}$ as entropy regularization instead of the differential entropy used in our paper. We noted the difference between the problem we are considering and the one considered in [5] in our paper, see lines 91-92 and corresponding footnote.
>
> **(2) The Gaussian mixture approach is likely to work only in low dimensions. It would be interesting to see when it fails, e.g. using the benchmark above.**
>
> As we explained in the previous answer the benchmark provided in [5] is not relevant for us as it considers another UEOT problem.
>
> **(3) The authors state that for OT-FM and UOT-FM in the FFHQ dataset, they use a 2-layer feed-forward network with 512 hidden neurons and ReLU activation. Where does this parameterization come from? It seems to be relatively small for a flow matching architecture on images, and does not seem to be the architecture used in the original papers.**
>
> It is important to understand here that we run our experiments in the latent space of the ALAE autoencoder, and not on the images directly. Accordingly, we adapted the architectures of the neural networks used in OT-FM and UOT-FM to work with latent codes. In this case, architectures such as fully connected neural networks are relevant.
>
> **(4) Why are FID scores not reported for the image translation tasks?**
>
> We conduct the image translation experiment in the latent space of ALAE autoencoder. For this reason, we did not report the FID metrics assessing the quality of the generated images but rather focus on assessing the quality of the generated latents and focus on the Frechet distance (FD) defined as the difference in means and covariances of the generated and target latents.
>
> However, to fully address the raised question, we report the FID scores between the generated images (produced by ALAE decoder from the generated latent codes) and target images distributions in the Table below. The results show that FID is nearly the same for all of the models under consideration. It supports our intuition that FID is indeed not a representative metric for assessing the performance of models performing the translation of latent codes.
>
> | 10k updates | ULight-OT (ours) | Light-SB [1]      | UOT-FM [2]           | Yang et al. [3]      | Choi et al. [4]   |
> |-------------|------------------|-------------------|----------------------|----------------------|-------------------|
> | FID         | $0.331 \pm 0.03$ | $0.331 \pm 0.03$  | $0.331 \pm 0.04 $  | $0.344 \pm 0.04 $  | $0.339 \pm 0.03 $ |
>
> **Concluding remarks**. Please respond to our post to let us know if the clarifications above suitably address your concerns about our work. We are happy to address any remaining points during the discussion phase; if the responses above are sufficient, we kindly ask that you consider raising your score.
>
> **References**
>
> [1] Korotin et al. "Light schrodinger bridge", ICLR 2024.
>
> [2] L. Eyring et al. Unbalancedness in neural monge maps improves unpaired domain translation. ICLR, 2024
>
> [3] J. Choi et al. Generative modeling through the semi-dual formulation of unbalanced optimal transport. arXiv preprint arXiv:2305.14777, 2023.
>
> [4] K. D. Yang et al. Scalable unbalanced optimal transport using generative adversarial networks. ICLR, 2018.
>
> [5] H. Janati et al. Entropic optimal transport between unbalanced gaussian measures has a closed form. NeurIPS, 2020.

---

> > ### Comment · Reviewer_t9a3 · 2024-08-09
> >
> > I thank the reviewers for their clarifications, and apologise for having missed this difference explained in lines 91-92. Thus, I increase my score to 6.

---

### Official Review · Reviewer_nig3 · 2024-07-13

**Soundness:** 3
**Presentation:** 3
**Contribution:** 2
**Rating:** 5
**Confidence:** 3

**Summary:**

This paper presents a lightweight solver for Unbalanced Entropic Optimal Transport (UEOT) that does not rely on neural network parametrization. Instead, the authors parameterize the potential functions of UEOT using Gaussian Mixture Models (GMM). This parametrization enables the derivation of a tractable joint coupling. By incorporating the parameterized potential into the dual objective, the authors achieve a simple and tractable loss function. Additionally, the paper provides a universal approximation result for GMM parametrization. Experiments are conducted on toy data (GMM) and image-to-image (I2I) translation.

**Strengths:**

- The paper proposes a simple and fast UEOT algorithm.
- The paper justifies the GMM parametrization by presenting generalization bounds.
- The paper demonstrates applicability to large-scale tasks such as I2I translation when combined with an autoencoder (AE).
- The paper is well-written, clear, and easy to follow.

**Weaknesses:**

- The method of parameterizing the potential function using GMMs was already proposed in LightSB [1]. The only change here is the switch to a UOT objective, making the methodological contribution minimal. Aside from the universal approximation result, the theoretical contributions are also limited.

- The experiments are not comprehensive. First, the experiments are conducted only on face-related data. More diverse datasets should be included. Second, the fairness of the comparisons is questionable. In the I2I experiments, the authors use the ALAE autoencoder, while some of comparison methods are implemented directly in the image space. All of the comparisons should be implemented in the latent space for fairness. Third, since U-LightOT are implemented on a latent space that captures attributes well, the attribute accuracy is expected to be high. Other than accuracy, more general metrics such as c-FID or FID should be used for comparison. Fourth, there is a lack of ablation studies on the number of Gaussian modals $N$ and $M$. This is very important and expected to be sensitive hyperparameter, thus, I believe authors should provide ablation studies on this parameter. Overall, the practical utility of the approach is questionable.

[1] Light Schrodinger Bridge, ICLR, 2024.

**Questions:**

- How does the performance change when parameterizing very high-dimensional and multi-modal GMM data with fewer or more $N,M$?
- In the I2I experiments, how does the number of modes in the GMM affect performance?
- In toy data experiments, does U-LightOT has lower transport plan costs and smaller Wasserstein distances between the target and generated distributions compared to other comparisons?

**Limitations:**

Discussed in Weakness section.

---

> ### Author Rebuttal · Authors · 2024-08-07
>
> Thank you for your thorough feedback. Please find the answers to your questions below.
>
> **(1) The method of parameterizing the potential function using GMMs was already proposed in LightSB [1]. The only change here is the switch to a UOT objective, making the methodological contribution minimal.**
>
> Our solver can be considered as the generalization of the one from [LightSB, 1] in the sense that it subsumes LightSB for the specific choice of $f$-divergences. However, this generalization is not straightforward or direct since our objective is built on the completely different principles:
>
> 1. Our solver is derived from minimizing $D_{\text{KL}}$ divergence (defined as a discrepancy between *positive measures*) between ground truth plan $\gamma^*$ and its approximation $\gamma_{\theta,\omega}$. This definition of divergence notably differs from the ordinary definition of $D_{\text{KL}}$ for **probability measures** used in [1].
>
> 2. We parametrize the entire plan using Gaussian mixtures while in [1] it is done only for conditional plans. It is an important difference, since in an unbalanced case the marginals of optimal plan do not coincide with source and target measures. Our parametrization allows sampling from the left marginal of UEOT plan and identifying potential outliers in the source measure.
>
> **(2) Aside from the universal approximation result, the theoretical contributions are also limited.**
>
> We partially agree with the reviewer that the proof of our Universal Approximation Theorem (UAT) is the most difficult and tricky among the results obtained in our paper. However, the theoretical contributions of our paper are not limited to this theorem. Our other results include:
> (1) Theorem 4.1 $-$ the derivation of the tractable optimization objective in terms of the $D_{\text{KL}}$ divergence between positive measures; (2) Proposition 4.2 $-$ the derivation of the bound for the estimation error of our solver; (3) Theorem A.4 $-$ derivation of the dual form of UEOT problem with the potentials belonging to the space $C_{2,b}(x)$ of continuous functions, bounded by the quadratic polynom (from the both sides) and additionally bounded by constant from above. Proof of each of these results is non-trivial and requires highly specialized knowledge in the diverse fields of mathematics and statistics.
>
> **(3) In the I2I experiments, the authors use the ALAE autoencoder, while some of comparison methods are implemented directly in the image space. All of the comparisons should be implemented in the latent space for fairness.**
>
> All of the methods included in comparison in the image-to-image translation task were *implemented in the latent space* of the ALAE autoencoder which is mentioned in the paper, see Section 5.2, line 276. We agree that it might written more clearly and will additionally emphasize this aspect in the final version of our paper.
>
> **(4) Since U-LightOT are implemented on a latent space that captures attributes well, the attribute accuracy is expected to be high. Other than accuracy, more general metrics such as c-FID or FID should be used for comparison.**
>
> We conduct the image translation experiment in the latent space of ALAE autoencoder. For this reason, we did not report the FID metrics assessing the quality of the generated images but rather focus on assessing the quality of the generated latents and focus on the Frechet distance (FD) defined as the difference in means and covariances of the generated and target latents.
>
> However, to fully address the raised question, we report the FID scores between the generated images (produced by ALAE decoder from the generated latent codes) and target images distributions in the Table below (Adult → Young translation). The results show that FID is nearly the same for all of the models under consideration. It supports our intuition that FID is indeed not a representative metric for assessing the performance of models performing the translation of latent codes.
>
> | 10k updates | ULight-OT (ours) | Light-SB [1]      | UOT-FM [2]           | Yang et al. [3]      | Choi et al. [4]   |
> |-------------|------------------|-------------------|----------------------|----------------------|-------------------|
> | FID         | $0.331 \pm 0.03$ | $0.331 \pm 0.03$  | $0.331 \pm 0.04 $  | $0.344 \pm 0.04 $  | $0.339 \pm 0.03 $ |
>
> **(5a) Lack of ablation studies on the number of Gaussian modes N and M.***
>
> To address the reviewer's concern, we perform additional experiments with our solver varying the number of Gaussian modes ($N$, $M$) in potentials.
>
> *Gaussians mixtures.* The setup of this experiment follows the setup introduced in our Section 5.1. We test our solver with diverse number of potentials $N\in\{1,3,5\}$ and  $M\in\{1,2,3,4,5\}$. The results are visualized in Fig. 1 of the **attached PDF file**. It can be seen that for insufficient number of modes in potentials, the solver exhibits issues with convergence and do not correctly solve the task.
>
> **(5b) How does the performance change when parameterizing very high-dimensional and multi-modal GMM data with fewer or more N and M?**
>
> Unfortunately, to assess the performance of our solver in such an experiment with multi-model GMM data, we need to have some kind of ground-truth solutions. However, for the multi-modal GMM data the solutions *are not available* making it hard to perform such an experiment. Following your comment, we qualitatively demonstrated the performance of our solver with varying number of potential modes $N,M$ for 2-dimensional Gaussian mixtures and quantitavely assess its performance in Image-to-Image translation task, see the answer above.
>
> **Concluding remarks**. Please respond to our post to let us know if the clarifications above suitably address your concerns about our work. We are happy to address any remaining points during the discussion phase; if the responses above are sufficient, we kindly ask that you consider raising your score.

---

> > ### Author Response · Authors · 2024-08-07
> >
> > **References.**
> >
> > [1] Korotin et al. "Light schrodinger bridge", ICLR 2024.
> >
> > [2] L. Eyring et al. Unbalancedness in neural monge maps improves unpaired domain translation. ICLR, 2024
> >
> > [3] K. D. Yang et al. Scalable unbalanced optimal transport using generative adversarial networks. ICLR, 2018.
> >
> > [4] J. Choi et al. Generative modeling through the semi-dual formulation of unbalanced optimal transport. NeurIPS, 2023

---

> > > ### Author Response · Authors · 2024-08-09
> > > **Further clarifications (1)**
> > >
> > > **In toy data experiments, does U-LightOT has lower transport plan costs and smaller Wasserstein distances between the target and generated distributions compared to other comparisons?**
> > >
> > > To answer this question, we compared our solver with different unbalancedness parameters $\tau\in\{1,10,50,100\}$ and LightSB for an experiment with a mixture of Gaussians. The results are presented in the table below. Note that our solver is designed to solve an unbalanced EOT problem with relaxed boundary conditions. This entails two properties. Firstly, our solver better preserves the properties of the input objects - indeed, it allows for the domain translation which preserves object classes even in the case of class imbalance. Secondly, due to the relaxed boundary condition for the target distribution, the distribution generated by our solver is naturally less similar to the target distribution than for balanced methods.
> > >
> > > The above intuitive reasoning is confirmed by the metrics we obtained. Indeed, as the $\tau$ parameter increases, when our method becomes more and more similar to balanced approaches, the normalized OT cost ($\mathbb{E}\_{x\sim p} \mathbb{E}\_{y\sim \gamma(y|x)} \frac{(x-y)^2}{2}$) between the source and generated distributions increases, and the Wasserstein distance between mapped $p$ and target distribution $q$ decreases. This property of our solver was noted in our paper, see Appendix C. LightSB [1] baseline, which is a purely balanced approach, shows the best quality in terms of Wasserstein distance and the worst in terms of OT cost.
> > >
> > > |                         | LightSB | U-LightOT (ours, $\tau=100$) |  U-LightOT (ours,$\tau=50$) | U-LightOT (ours,$\tau=10$) | U-LightOT (ours,$\tau=1$) |
> > > |-------------------------|---------|----------------------|---------------------|--------------------|------|
> > > | OT cost        | 3.952         | 3.931   | 3.874                | 2.913               | 2.023              |
> > > | $\mathbb{W}_2$-distance | 0.088| 0.091   | 0.138                |      1.107          | 2.044             |
> > >
> > > It is important to note that our method offers a flexible way to select a domain translation configuration that allows for better preserving the properties of the original objects or generating a distribution closer to the target one. The final optimal configuration selection remains at the discretion of the user. At the same time, balanced approaches do not allow making a choice in favor of preserving the properties of the original objects.

---

> > > > ### Author Response · Authors · 2024-08-09
> > > > **Further clarifications (2)**
> > > >
> > > > **(a) In the I2I experiments, how does the number of modes in the GMM affect performance?**
> > > >
> > > > **(b) ...since U-LightOT are implemented on a latent space that captures attributes well, the attribute accuracy is expected to be high**
> > > >
> > > > Up to the request, we perfom an ablation study of our U-LightOT solver with different number on Gaussian components in potentials $N,M$. Similarly to Appendix C of our paper, we run the solver in $\textit{Young}\rightarrow\textit{Adult}$ translation with 3K steps, $\varepsilon=0.1$ and set $\tau=100$. The quantitative results (accuracy of keeping the class, accuracy of mapping to the target, FD of generated latents vs target latents) are presented in the Tables below.
> > > >
> > > > *FD of generated latents (less is better)*
> > > > |  N/M   | 1            | 2            | 4            | 8            | 16           | 32           | 64           |
> > > > |-----|--------------|--------------|--------------|--------------|--------------|--------------|--------------|
> > > > | 1   | 29.18 ± 0.05 | 31.43 ± 2.83 | 31.49 ± 1.65 | 31.66 ± 2.36 | 32.59 ± 2.12 | 31.11 ± 2.65 | 33.60 ± 3.40 |
> > > > | 2   | 29.38 ± 0.70 | 27.75 ± 0.12 | 27.98 ± 0.30 | 30.15 ± 2.33 | 31.61 ± 2.65 | 30.46 ± 2.71 | 29.60 ± 1.29 |
> > > > | 4   | 29.44 ± 1.05 | 28.63 ± 2.18 | 28.78 ± 1.06 | 28.17 ± 1.35 | 28.95 ± 2.21 | 27.94 ± 1.18 | 30.29 ± 2.72 |
> > > > | 8   | 30.79 ± 2.67 | 28.55 ± 0.75 | 29.54 ± 3.23 | 27.82 ± 0.89 | 27.10 ± 0.23 | 29.98 ± 0.50 | 27.57 ± 1.03 |
> > > > | 16  | 27.15 ± 0.34 | 30.95 ± 1.20 | 27.69 ± 0.66 | 28.71 ± 2.17 | 28.19 ± 0.50 | 27.58 ± 0.78 | 28.31 ± 0.92 |
> > > > | 32  | 28.86 ± 1.35 | 30.64 ± 4.05 | 27.36 ± 0.85 | 28.01 ± 0.87 | 27.99 ± 1.59 | 29.52 ± 1.65 | 29.14 ± 1.91 |
> > > > | 64  | 30.08 ± 3.63 | 29.12 ± 1.58 | 30.07 ± 2.67 | 30.04 ± 1.05 | 29.15 ± 2.00 | 29.48 ± 2.43 | 29.08 ± 1.53 |
> > > >
> > > >
> > > > *Accuracy of keeping the class (less is better)*
> > > >
> > > > |  N/M   | 1            | 2            | 4            | 8            | 16           | 32           | 64           |
> > > > |-----|--------------|--------------|--------------|--------------|--------------|--------------|--------------|
> > > > | 1   | 87.78 ± 0.62 | 88.11 ± 0.69 | 88.50 ± 0.27 | 88.38 ± 0.30 | 87.97 ± 0.38 | 88.19 ± 0.49 | 88.45 ± 0.37 |
> > > > | 2   | 88.65 ± 0.25 | 88.57 ± 0.64 | 87.54 ± 1.08 | 88.62 ± 0.75 | 88.09 ± 0.44 | 88.06 ± 0.39 | 88.75 ± 0.51 |
> > > > | 4   | 87.89 ± 0.44 | 87.84 ± 0.39 | 88.22 ± 0.87 | 87.82 ± 0.64 | 88.85 ± 0.70 | 87.25 ± 0.43 | 88.09 ± 0.66 |
> > > > | 8   | 88.54 ± 0.95 | 88.52 ± 0.48 | 88.29 ± 0.40 | 88.27 ± 0.37 | 87.93 ± 0.73 | 88.88 ± 0.78 | 87.56 ± 0.75 |
> > > > | 16  | 88.38 ± 0.73 | 88.89 ± 0.50 | 88.02 ± 0.45 | 88.19 ± 0.32 | 87.80 ± 0.70 | 87.84 ± 0.62 | 87.94 ± 0.57 |
> > > > | 32  | 87.97 ± 0.75 | 88.87 ± 0.59 | 86.99 ± 0.16 | 87.71 ± 0.50 | 87.50 ± 0.44 | 87.71 ± 0.76 | 88.08 ± 0.37 |
> > > > | 64  | 87.13 ± 0.73 | 88.23 ± 0.58 | 87.70 ± 0.91 | 87.56 ± 0.52 | 87.99 ± 0.48 | 88.83 ± 0.43 | 88.30 ± 0.59 |
> > > >
> > > >
> > > > *Accuracy of mapping to the target (less is better)*
> > > >
> > > > |  N/M   | 1            | 2            | 4            | 8            | 16           | 32           | 64           |
> > > > |-----|--------------|--------------|--------------|--------------|--------------|--------------|--------------|
> > > > | 1   | 79.09 ± 0.02 | 79.48 ± 0.17 | 79.09 ± 0.72 | 78.85 ± 0.41 | 78.86 ± 0.04 | 79.02 ± 0.31 | 78.00 ± 0.34 |
> > > > | 2   | 79.26 ± 0.52 | 79.10 ± 0.66 | 78.95 ± 0.49 | 79.68 ± 0.46 | 79.65 ± 0.29 | 79.44 ± 0.74 | 79.49 ± 0.42 |
> > > > | 4   | 78.80 ± 0.71 | 79.27 ± 0.64 | 79.68 ± 0.31 | 79.84 ± 0.50 | 79.16 ± 0.47 | 79.57 ± 0.92 | 78.88 ± 1.34 |
> > > > | 8   | 78.41 ± 0.16 | 79.36 ± 0.54 | 78.74 ± 0.20 | 78.80 ± 0.69 | 79.11 ± 0.89 | 78.43 ± 1.01 | 79.06 ± 0.71 |
> > > > | 16  | 79.44 ± 0.82 | 79.27 ± 0.40 | 79.21 ± 1.15 | 79.47 ± 0.57 | 79.35 ± 0.63 | 79.70 ± 0.67 | 78.34 ± 1.15 |
> > > > | 32  | 78.59 ± 0.42 | 79.55 ± 0.59 | 79.27 ± 0.79 | 78.70 ± 1.09 | 79.37 ± 0.63 | 78.68 ± 0.77 | 79.42 ± 0.77 |
> > > > | 64  | 79.11 ± 1.02 | 78.41 ± 0.66 | 79.29 ± 0.36 | 77.91 ± 0.45 | 78.19 ± 1.18 | 80.27 ± 0.96 | 79.60 ± 0.47 |
> > > >
> > > >
> > > > The results show that in the considered task, our solver provides good performance even for small number of Gaussian components. This can be explained by the smoothness of the latent representations of data in ALAE autoencoder.

---

> > > > > ### Comment · Reviewer_nig3 · 2024-08-11
> > > > >
> > > > > I appreciate the authors for their clarifications, particularly regarding the comparison between this work and [1]. Moreover, I thank the authors for the extensive experiments conducted. Based on this, I would like to raise my score to 5.

---

### Author Rebuttal · Authors · 2024-08-07

Dear reviewers,

thank you for your thorough and detailed reviews! We are highly inspired by the fact that you agree on the importance of our theoretical results (Reviewer bWBu, vYvs), clarity (Reviewer WAcu, bWBu, t9a3, nig3) of our paper and mark the efficiency of the our solver (Reviewer bWBu, vYvs). We hope that our U-LightOT algorithm would be easy to use in practical applications.

We will incorporate the changes suggested by the reviewers in the final version of our paper. We list the changes below:

(a) **Main text** $-$ addition of the Table with wall-clock comparison of our U-LightOT solver and its competitors (Reviewer ) plus minor requested clarifications,

(b) **Additional experiment in Appendix C** section $-$ ablation study of our solver with different number of gaussian modes in potentials(**Reviewers  nig3, bWBu**),

(c) **Additional experiment in Appendix E** section $-$ *Gaussian Mixtures with outliers* experiment showing the robustness of our solver towards potential outliers (**Reviewer bWBu**).

Please find Figures for experiments requested by the reviewers  nig3, bWBu in the **attached PDF file**.

Please find the answers to your questions below.

---

> ### Author Response · Authors · 2024-08-14
> **Concluding remarks**
>
> Since the end of the rebuttal period is approaching, we want to thank the reviewers for their time spent on their reviews and subsequent discussion. We are grateful for your interesting and valuable suggestions and will add the changes to the final version of our paper. In addition to the changes listed in the [general comment](https://openreview.net/forum?id=co8KZws1YK&noteId=CR3zz8IfwE), we will include
>
> 1. (Addition to **Appendix C**) Table with OT cost between the source and generated distributions and Wasserstein-2 distance between the generated and target distributions in Gaussian mixture experiment (with varying parameter $\tau$);
> 2. (Addition to **Appendix E**) Tables with additional metrics for comparing our method and its competitors (accuracy, FD, FID between the generated latents and the target ones).

---

### Decision · Program_Chairs · 2024-09-25

**Decision:**

Accept (poster)

**Comment:**

The paper introduces a new solver for Unbalanced Entropic Optimal Transport, using a novel parameterization of the dual potential. The reviewers acknowledged that the paper offers theoretical insights into the approximation capabilities of the proposed method and that the authors made a significant effort in the rebuttal to improve some of the numerical experiments. However, all reviewers expressed concerns that the contributions are too incremental, primarily combining existing methodological and theoretical techniques from [39] (and [30], which is the same paper) and relying on the same set of experiments.

The introduction of the paper does not clearly outline the connection to [39] or explain the differences, failing to adequately motivate the relevance of the contribution. As a result, the paper does not clearly convey what makes this generalization to an unbalanced solver both challenging and important. Furthermore, the numerical evaluation of the proposed method’s improvements is mostly qualitative, without new compelling experiments to demonstrate the practical benefits from an applied perspective. Instead of relying on the same set of experiments as in [39], there should have been more innovation in this area, with stronger benchmarking to better highlight the advantages of the method.

For these reasons, I recommend rejecting this paper.